# Remote and local drivers of oxygen and nitrate variability in the shallow oxygen minimum zone off Mauritania in June 2014

Soeren Thomsen[1,2], Johannes Karstensen[2], Rainer Kiko[2], Gerd Krahmann[2], Marcus Dengler[2], and Anja Engel[2]

[1]LOCEAN-IPSL, IRD/CNRS/Sorbonnes Universites (UPMC)/MNHN, UMR 7159, Paris, France
[2]GEOMAR - Helmholtz Centre for Ocean Research Kiel, Düsternbrooker Weg 20, 24105 Kiel, Germany

*Correspondence to:* Soeren Thomsen, soeren.thomsen@locean-ipsl.upmc.fr

**Abstract.** Upwelling systems play a key role in the global carbon and nitrogen cycles and are also of local relevance due to their high productivity and fish resources. To capture and understand the high spatial and temporal variability of physical and biogeochemical parameters found in these regions novel measurement technics have to be combined in an interdisciplinary manner. Here we use high-resolution glider-based physical-biogeochemical observations in combination with ship-based un-
derwater vision profiler, sensor and bottle data to investigate the drivers of oxygen and nitrate variability across the shelf break off Mauritania in June 2014. Distinct oxygen and nitrate variability shows up in our glider data. High oxygen and low nitrate anomalies were clearly related to water mass variability and probably linked to ocean transport. Low oxygen and high nitrate patches co-occurred with enhanced turbidity signals close to the seabed, which suggests locally high microbial respiration of resuspended organic matter near the sea floor. This interpretation is supported by high particle abundance observed by the
underwater vision profiler and enhanced particle-based respiration rate estimates close to the seabed. Discrete in-situ measurements of dissolved organic carbon and amino acids suggest the formation of dissolved organic carbon due to particle dissolution near the seabed fueling additional microbial respiration. During June an increase of the oxygen concentration on shelf-break of about 15 $\mu$mol kg$^{-1}$ is observed. These changes go along with meridional circulation changes but cannot be explained by typical water mass property changes. Thus our high-resolution interdisciplinary observations highlight the complex interplay
of remote and local physical-biogeochemical drivers of oxygen and nitrate variability off Mauritania, which cannot be captured by classical shipboard observations alone.

## 1   Introduction

The Mauritanian upwelling region is located in the shadow zone of the eastern tropical North Atlantic (ETNA), an area characterized by sluggish mean circulation (Luyten et al., 1983). The local balance between oxygen supply (ventilation) and res-
piration creates a vertical oxygen structure with two minima indicating a shallow and a deep Oxygen Minimum Zone (OMZ) (Karstensen et al., 2008; Brandt et al., 2015). The deep OMZ below has a core depth of about 400 m and minimum oxygen concentrations around 40 $\mu$mol kg$^{-1}$ (Karstensen et al., 2008; Brandt et al., 2015). Here the focus lies on the shallow OMZ with a core depth of about 100 m and oxygen concentrations between 40 and 60 $\mu$mol kg$^{-1}$ (Karstensen et al., 2008; Brandt

et al., 2015; Klenz et al., 2018). Close to the coast near the seabed high oxygen variability and even oxygen concentrations well below $35\,\mu\mathrm{mol\,kg}^{-1}$ have been reported for the shallow OMZ at 18°N (Yücel et al., 2015; Gier et al., 2017). Despite the potential importance of the shallow OMZ for the ecosystem in the ETNA and off Mauritania it has gained much less attention compared to the deep OMZ so far (Brandt et al., 2015). Although recent observational and modeling studies investigated i.e.

the importance of low oxygen submesoscale coherent vortices, also called "dead zone eddies" (Karstensen et al., 2015), for the maintenance of the offshore shallow OMZ (Schütte et al., 2016; Frenger et al., 2018), detailed studies on the shallow OMZ are sparse, particularly close to the Mauritanian coast. Reproducing the distinct vertical structure i.e. the separation between the shallow and deep OMZ in the ETNA is, even in high resolution (1/10°) coupled physical-biogeochemical model simulations, a problem (Duteil et al., 2014; Frenger et al., 2018) and calls for a better understanding of remote and local physical and

biogeochemical drivers of oxygen distribution and variability off Mauritania.

The eastern boundary region plays a key role in the characterization of the ETNA OMZ. In particular the structure of the coastal wind stress forcing and the variability of the boundary current system are of importance. The eastern boundary circulation is dominated by poleward flow. Historically, two poleward flows were identified, the surface intensified Mauritanian Current (Kirichek, 1971) and the subsurface Poleward Undercurrent (Mittelstaedt and Hamann, 1981). However, recent obser-

vations (Klenz et al., 2018), eastern boundary circulation theory (Fennel, 1999) and a model study (Kounta et al., 2018) suggest that both flows are expressions of the same forcing and should thus be named Mauritanian Current (Klenz et al., 2018) or West African Poleward Boundary Current (Kounta et al., 2018). The currents vary in volume and property transport with a clear seasonal signal that aligns with the coastal wind stress curl variability. During boreal winter between January and April the Intertropical Convergence Zone (ITCZ) migrates southwards and alongshore winds drive the maximum upwelling in the study

area near 18°N (Mittelstaedt, 1983; Van Camp et al., 1991; Lathuilière et al., 2008). During this period cold and nutrient-rich waters that ultimately stem from the South Atlantic are being upwelled and enhance primary productivity (Mittelstaedt, 1983, 1991; Peña-Izquierdo et al., 2015). During May and July the upwelling ceases and a rapid increase in surface temperature is observed (Mittelstaedt, 1983; Lathuilière et al., 2008). A recent modeling study by Kounta et al. (2018) describes the seasonality of the near surface circulation by low-frequency coastal trapped wave activity generated by seasonal wind fluctuations

along the African shores and identified a semi-annual cycle with transport maxima in spring and fall. Additionally the authors give an in depth overview over the historical and more recent literature and discussion on the West African near surface current system. Klenz et al. (2018) describe the seasonality of the boundary current system based on ship-board velocity measurements of multiple cruises. During the upwelling season (January to April) the authors found a weak poleward flow but an equatorward coastal jet whereas a surface intensified poleward flow with velocities well above 30 cm/s was observed during the relaxation

period (May - July). They attribute the seasonal differences of the eastern boundary flow to local Sverdrup dynamics.

Three main water masses are present in the upper 300 m of the water column off Mauritania (Tomczak, 1981; Tsuchiya et al., 1992; Stramma et al., 2005): near the surface Subtropical Underwater (STUW) is found at densities below $25.8\,\mathrm{kg\,m}^{-3}$ (Stramma et al., 2005). The STUW originates from the mostly wind driven subduction of high salinity waters. The subduction takes place over a rather wide area and can even be seasonal, creating a water mass with variable temperature/salinity (T/S) characteristics

and high oxygen concentrations (Stramma et al., 2005; Karstensen et al., 2008). Below the STUW the water mass characteris-

tic is bounded by North Atlantic (NACW) and South Atlantic Central Water (SACW) emphasizing the promiment role of the South Atlantic in ventilating the North Atlantic OMZ region (Karstensen et al., 2008). SACW is, along isopycnals, less saline and colder than the NACW. In a strict sense the SACW is not fully equal to its southern hemisphere source because of the modifications due to mixing during its spreading into the northern hemisphere. In T/S space a clear separation between NACW

and SACW is seen for waters less dense than 26.8 $kg\,m^{-3}$ isopycnal (Rhein et al., 2005; Kirchner et al., 2009). This so called "upper" central water mass density range is also of interest here but for simplicity we just refer to NACW and SACW. Besides warm and saline, the NACW is characterized by higher oxygen and lower nutrient (nitrate, phosphate) concentrations (Poole and Tomczak, 1999; Peña-Izquierdo et al., 2012). In contrast the cold and fresh SACW has lower oxygen and higher nutrient concentrations (Poole and Tomczak, 1999). Interestingly, this picture looks different when coming closer to the Mauritanian

coast and at 18 °N: here the SACW is transported in the Poleward Undercurrent and shows up as a local oxygen maximum but which is simply because the surrounding waters are a mixture of old NACW and SACW and thus have low oxygen content (and high nutrients) (Peña-Izquierdo et al., 2012; Brandt et al., 2015). Besides the local water column remineralization and respiration, the benthic oxygen uptake (Dale et al., 2014) and nitrogen loss processes (Sokoll et al., 2016) might also contribute to shaping the oxygen and nutrient distributions and variability off Mauritania.

In the context of the biological reduction of oxygen in the water column the microbial respiration plays a key role and is linked to the regionally varying production and export of organic matter. Oxygen respiration rates decrease with depth and typical rate estimates for the deep OMZ are about 5 $\mu mol\,kg^{-1}\,year^{-1}$ at about 500 m and increase to about 10 $\mu mol\,kg^{-1}\,year^{-1}$ at 100 m depth (Karstensen et al., 2008; Engel et al., 2017). However, in the high productive areas close to the coast off Mauritianan strong nutrient upwelling stimulates high primary production and leads to the accumulation of organic matter in

particulate (POC) (Fischer et al., 2009; Iversen et al., 2010) and dissolved (DOC) form. Due to this high input of organic matter local respiration close to the coast can be expected to be much more relevant for the instantaneous oxygen distribution and variability. Brandt et al. (2015) estimated a diapycnal oxygen supply term of a few $\mu mol\,kg^{-1}\,day^{-1}$ off Mauritania pointing to a very fast remineralization processes which maintain the strong vertical oxygen gradient in the presence of strong vertical mixing. Likewise in high productive "dead zone eddies" respiration rates of more than 0.1 $\mu mol\,kg^{-1}\,day^{-1}$ have been

reported (Karstensen et al., 2015) emphasizing the respiration potential in regions with high organic matter abundance. Due to the direct interlinkage between respiration and organic matter remineralization we also investigate the role of local organic matter dynamics off Mauritania for the observed oxygen and nitrate variability.

    While part of the POC is exported to the deeper ocean via gravitational sinking of larger particles, DOC produced by extracellular release, cell lysis and break-up, or by enzymatic dissolution of particles, keeps organic carbon in surface waters,

where it becomes available for microbial uptake and respiration. On the seafloor, particle degradation by microorganisms and benthic fauna also releases high amounts of DOC (Loginova et al., 2016). Microbial respiration is the main biogenic sink of oxygen in the ocean (Azam et al., 1983). How much DOC is respired and contributes to the marine oxygen sink compared to particle associated respiration is largely unknown and depends on physical mixing, on the chemical quality of the organic components and the respiration activity of microbes. Accumulation of DOC in Atlantic surface waters may be controlled by

the availability of new nutrients and is assumed to be especially high in upwelling regions (Romera-Castillo et al., 2016). In

the upwelling regions off Northwest Africa and North-west Iberia, production rates of DOC and POC were found to be similar (Alvarez-Salgado et al., 1999, 2007). Information on concentration, chemical composition and associated lability of DOM in the ETNA, and in EBUS in general, is however scarce. DOC is a heterogeneous pool of organic compounds, often categorized by its turnover time into labile (hours-days), semi-labile (weeks-months) and refractory (years-centuries) components. Semi-labile DOC is mainly represented by high molecular weight DOM, i.e. biopolymers, such as combined carbohydrates and hydrolysable amino acids (Benner, 2002). Semi-labile and refractory DOC thus reside long enough in seawater to be transported away from their source of production by ocean currents. For the ETNA upwelling regions off North-Africa, offshore transport of DOC has been hypothesized to support microbial respiration in the more oligotrophic open Atlantic regions (Alvarez-Salgado et al., 2007). Understanding the chemical composition, offshore transport and microbial respiration of fresh DOC in the highly productive ETNA is thus important to understand the remote and local drivers of oxygen and nitrate variability.

As it is typical for an EBUS, the Mauritanian upwelling region has been found highly variable in its local physical and biogeochemical characteristics, both in time and space (Schafstall et al., 2010; Peña-Izquierdo et al., 2012; Yücel et al., 2015). In this study we investigate the oxygen and nutrient (here the sum of nitrate and nitrite) distribution along 18°N and in the depth range of the upper OMZ at the end of the upwelling season in June 2014, during the transition phase from strong to weaker upwelling. We aim to better understand the drivers of oxygen and nitrate variability to evaluate and possibly improve regional model simulations and predictions of the future state off the Mauritanian oxygen minimum zone and the associated ecosystem. Our study is mainly based on a data set that includes physical (velocity, temperature, salinity) and biogeochemical (oxygen, nitrate, turbidity and chlorophyll) parameters measured by sensors attached to an autonomous underwater glider. Significant amount of data were acquired which allows to analyze the high variability of physical and biogeochemical parameters in the Mauritanian upwelling system in a very detailed view and with much better statistics then obtained from ship based observations. Our study also devotes attention to the role of the pelagic processes close to the benthos in contributing to the local oxygen and nitrate structure. Benthic lander observations with novel Lab-on-Chip technology reported high nirate and nitrite variability on timescales of less than 40 hours for our working area (Yücel et al., 2015). As underwater glider for the water column, the high resolution lander based time series observations highlight the advantage of new technology in revealing variability that can not be captured by traditional observing methods. However, ship-based profile and bottle data allowed us to carry out high precision reference data as well as to collect a suite of parameters so far not accessible from autonomous instrumentation.

Based on the observational data at hand we aim to decompose and identify drivers for remote (via transport) and local (biogeochemical cycling) variability in oxygen and nitrate. This includes estimating local microbial particle-associated oxygen respiration as well as the possible role of DOC as microbial substrate. The paper is structured as follows: In section 2 the observational datasets, including data processing and calibration procedures as well as methods are described. In section 2.6 and 2.7 we introduce two methods (i) the local Apparent Oxygen Utilization (AOU) method and (ii) the Extended Optimum Multiparameter (OMP) method, which we use to decompose the observed parameter fields into remote and local processes. In the first result section 3.1 the mean oceanic background conditions during June 2014 are described. This is followed by a description of the temporal variability of the along-shore circulation in June 2014 (subsection 3.2). The variability of oxygen

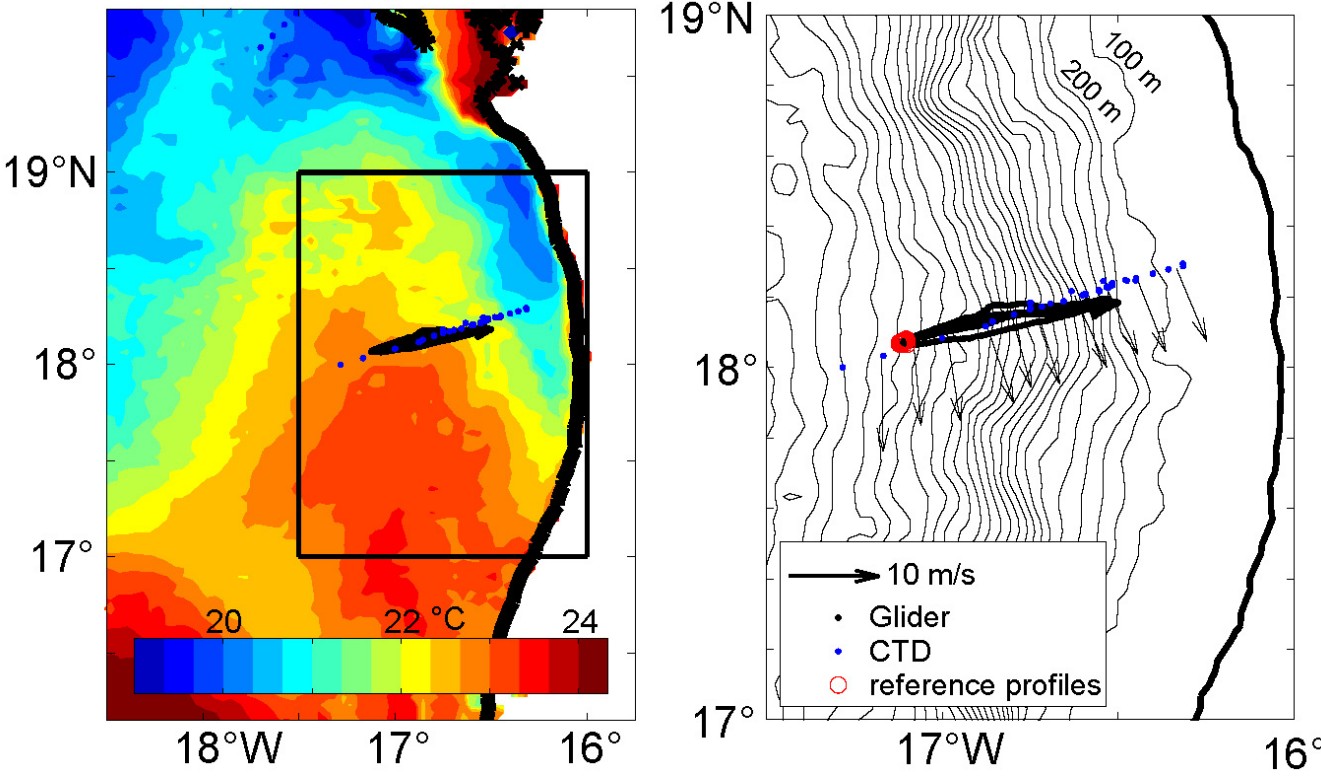

**Figure 1.** Sea surface temperature (SST) off Mauritania in June 2014 (left). The SST composite is based on in total 14 usable (few clouds) MODIS Aqua and Terra measurements (both day and night) at 8 days between 12 and 22 June 2014. Multiple SST measurements during a single day were averaged prior to the overall temporal average. The water depth (black contours), the glider tracks (black dots), offshore reference profiles (see section: 2.6, red circles), as well as conductivity, temperature and depth (CTD) stations (blue dots) are shown on the right. The wind direction and strength are shown with black arrows.

and nitrate is investigated in subsection 3.3 which includes the synoptic variability along single glider transects (subsection 3.3.1) and the low frequency changes during June 2014 (section 3.3.2). These analyses include the results of the local AOU and OMP analysis in regard to remote (transport) and local (respiration and remineralization) processes. The local remineralization signals are compared with oxygen respiration rates derived from particle abundance measurements with the underwater vision profiler UVP5 in section 3.4. In subsection (3.5) we investigate the possible role of DOM, including DOC and dissolved hydrolysable amino acids (DHAA), for the local respiration and remineralization processes. We discuss and conclude our results in section 4 and 5 respectively.

## 2 Observational datasets, data processing and methods

In June 2014 the research cruise Meteor 107 (M107) was carried out off Mauritania in the framework of the interdisciplinary collaborative research center SFB 754 "Climate-Biogeochemistry Interactions in the Tropical Ocean" funded by the Deutsche Forschungsgemeinschaft (DFG). The multi-platform experiment was carried out following a cross-shelf transect near 18°N (Fig. 1, e.g. Yücel et al. (2015); Gier et al. (2017)). In this study we analyze and interpret physical and biogeochemical glider-based measurements of one Slocum glider (IFM13) deployment in combination with ship-based profiles and bottle data.

### 2.1 Ship-based velocity, sensor and nutrient measurements

The ship-based instrumentation included a vessel-mounted acoustic Doppler current profiler and a pumped Seabird SBE 9-plus CTD system, equipped with double sensor packages for temperature, conductivity (salinity), and oxygen, and with single sensors for chlorophyll fluorescence and turbidity. In total 62 CTD stations were carried out along the 18°N transect between June 8 and 27, 2014. The CTD was mounted on a 24 bottle General Oceanic rosette system used to collect discrete water samples for analysis of salinity and various biogeochemical parameters. From the observational data we calculated conservative temperature ($\Theta$), absolute salinity ($S_A$), and potential density anomaly ($\sigma_\Theta$) using the Thermodynamic Equation of Seawater-2010 Matlab Toolbox Version 3.04 (McDougall and Barker, 2011). The salinity of bottle samples were measured on board RV Meteor with a Guildline Autosal 8 model 8400B salinometer and used to calibrate the CTD conductivity sensor. The calibrated salinity measurements have an accuracy of 0.002 g/kg. The CTD oxygen sensor (SBE 43) was calibrated with Winkler titration oxygen measurements of discrete water samples obtained from the rosette (Winkler, 1888; Grasshoff et al., 1983). Considering the scatter of the calibrated oxygen measurements an accuracy of 2.5 $\mu$mol kg$^{-1}$ from the oxygen concentrations was determined. In total 835 bottle samples along 18°N were analysed for this study for nitrate and nitrite using a QuAAtro autoanalyzer (Seal Analytical) on board. The nutrient data was used for sensor intercalibration (section 2.5) and to characterize the upper shelf nutrient distribution where not glider measurements were available. See also Yücel et al. (2015) and Fiedler et al. (2016) where parts of the nutrients dataset have already been published. The horizontal circulation was observed with two vessel-mounted acoustic Doppler current profilers (Ocean Surveyor with 38-kHz and 75-kHz). Here we focus on the upper 250 m of the water column and thus only make use of the 75-kHz instrument. The bin size was set to 8 m and the ping rates for both were 2–3 s. The first reliable depth bin is centered in 18 m depth. Water track and amplitude calibration was performed as described in Fischer et al. (2003).

### 2.2 Underwater Vision Profiler, particle abundance and particle-associated respiration rate estimation

An Underwater Vision Profile 5 (UVP5; Picheral et al. (2010)) was mounted on the General Oceanic rosette and operated during downcasts to obtain full depth particle size spectra (0.06 mm to 26.8 mm equivalent spherical diameter, ESD). Microbial particle-associated respiration rates (PARR) were calculated according to Kalvelage et al. (2015) for particle sizes between 0.06 and 10.64 mm ESD as a function of particle size (ESD, mm), ambient temperature and oxygen concentration. The empiric relationship of particle size and oxygen respiration ($r = 1.8417 \cdot ESD^{1.8}\ nmol\,h^{-1}$) is based on a dataset close to the study

area in the northern Mauritanian upwelling systems (Iversen et al., 2010). The relationship published by Kalvelage et al. (2015) well describes the lower bound of the data shown in Fig. 8c of Iversen et al. (2010). The upper bound can be well described by the relationship $r = 5.9691 \cdot ESD^{1.5378} \ nmol h^{-1}$ (M. Iversen, pers. communication). We use this relationship to highlight the potential upper bound of our PARR estimates. In both cases we use a correction coefficient of 2 for temperature differences and also correct for possible diffusion limitation of oxygen respiration at low environmental oxygen concentrations. Both corrections are explained in more detail in Kalvelage et al. (2015). Estimates based on the upper bound fit are generally 13- to 16-fold higher then those based on the lower bound fit. We have to expect a high variability in PARR, as the energy content of particles likely changes with depth, because of preferential use of easily degradable compounds. On the other hand, particles need to be colonized by microbes first to be efficiently degraded. The upper bounds of our estimates seem comparatively high and thus we base our discussion on the lower bound estimates following Kalvelage et al. (2015). The PARR of single size classes were multiplied by particle abundances in the different size classes and summed up for all size classes to obtain PARR.

## 2.3 Dissolved organic carbon

For DOC, samples (20 ml) were collected in duplicate, filtered through combusted (8 hrs, 500°C) GF/F filters and filled into combusted (8 hrs, 500°C) glass ampoules. Samples were acidified with 80 $\mu L$ of 85% phosphoric acid, heat sealed immediately, and stored at 4 °C in the dark until analysis. DOC samples were analyzed by high-temperature catalytic oxidation (TOC-VCSH, Shimadzu) modified from Sugimura and Suzuki (1988) and as described in more detail in Engel and Galgani (2016).

## 2.4 Dissolved hydrolysable amino acids

Dissolved hydrolysable amino acids (DHAA) were determined using a 1260 HPLC system (Agilent), following the methods described by Lindroth and Mopper (1979) and Dittmar et al. (2009), with modifications as described in Engel and Galgani (2016). Duplicate samples (5 ml) were filled into pre-combusted glass vials (8 hrs, 500°C) and stored at –20 °C until analysis. For DHAA, samples were first filtered through 0.45 $\mu m$ Millipore Acrodisc syringe filters. A measure for the diagenetic state of organic matter is the amino acid-based degradation index (DI) (Dauwe and Middelburg, 1998; Dauwe et al., 1999). For the calculation of DI from THAA in this study, mole percentages of amino acid were standardized using averages, and standard deviations and multiplied with factor coefficients as given in Dauwe et al. (1999) based on Principal Component Analysis. DI values often range between +2 and -2, with lower values indicating more degraded, higher values more fresh organic material. A total of 578 DOC and 347 DHAA samples were collected at 46 and 35 CTD stations along the 18°N transect in the upper 205 m of the water column (Fig. 1).

## 2.5 Glider-based measurements

The SLOCUM G2 underwater electric glider IFM13 was operated from June 12 to the June 27 2014 and did 6 sections of roughly 68 km each. The glider was programmed to dive to a maximum depth of 300 m. The glider was equipped with a pumped CTD and an Aanderaa optode was mounted on the tail of the gliders to measure concentrations of dissolved oxygen.

Data processing and calibration of salinity and oxygen measurements followed the procedures described and cited in Thomsen et al. (2016). The glider was equipped with a Satlantic Deep SUNA nutrient sensor measuring nitrate and nitrite, thereafter named $\sum NO_x$ following Yücel et al. (2015). The configuration and data processing of the SUNA was done as described in Karstensen et al. (2017). For the IFM13 deployment we linearly corrected the SUNA ($\sum NO_x$) measurements by the following

function $NO_x(calibrated) = 1.065 * NO_x(measured) + -0.083878$. This correction was determined by comparing the sum of nitrate and nitrite from the ships observation with the $\sum NO_x$ on isopycnals . This in-situ calibration reveals an accuracy (RMS) of 1.3 $\mu$mol kg$^{-1}$ and even lower values below 50 m water depth which is the focus of this study. The manufacturer gives a precision of the SUNA measurements of 0.3 $\mu$mol kg$^{-1}$ . The sensor calibration carried out by the manufacturer was used for the chlorophyll fluorescence and turbidity sensors (Wetlabs FLNTU) measurements. The chlorophyll fluorescence

measurements were converted into approximate chlorophyll a concentrations with the sensor specific scale factor provided by the manufacturer. IFM13 was equipped with an altimeter and the glider was programmed to turn at distance to the bottom of 20 m. Depth-averaged velocities can be constructed from gliders between two surfacings using the difference between the dead-reckoned and the observed position (Pietri et al., 2013, 2014; Thomsen et al., 2016). Here we make use of this to describe the temporal change of the horizontal circulation on the shelf break.

## 2.6   Local Apparent Oxygen Utilization (AOU) method

Given the long integration time for remineralization and respiration processes in OMZ regions the background signal in nutrients enrichment and oxygen loss is large. Thus locally generated signals can be difficult to detect. Here we use two different approaches to separate locally and remotely forced oxygen and nitrate variability. The first one is based on a determination of the local AOU ("AOU method") and the second approach makes use of a water mass mixing analysis ("OMP method").

AOU is defined as the difference between oxygen saturation calculated from an empirical relationship based on temperature and salinity (Weiss, 1970), and the observed oxygen. The "AOU method" is based on a simple model for an upwelling region that is controlled by along-isopycnal (lateral) spreading of the water masses from the offshore regions (remote signal) towards the coast and respiration/remineralization within the area from the reference to the coast (local signal). By identifying a "suitable" mean AOU profile in an offshore region the two components could easily be separated. In practice we fitted a 4th

order polynomial function to a group of 18 offshore AOU reference profiles using density as the independent variable (Fig. 1b, Fig. 2a, b). The so obtained AOU($\sigma$) function was applied to the observed density field to reconstruct the respective remote signal AOU field, which in turn was subtracted from the observed oxygen concentrations to obtain the local AOU anomaly. To minimize the impact of possible local ventilation we limit the analysis to the interior ocean below the mixed layer but also above the depth where NACW and SACW T/S characteristic are difficult to distinct. Moreover a smooth offshore AOU ($\sigma$) profile was needed to allow a reasonable fit. Thus our final solution space covers all waters between 26.1$< \sigma <$26.7 kg m$^{-3}$ .

profile was needed to allow a reasonable fit. Thus our final solution space covers all waters between 26.1$< \sigma <$26.7 kg m$^{-3}$ . We applied the same strategy to estimate a $\sum NO_x$ from the $\sigma$ field in order to estimate local remineralization on total nitrate and ultimately the local stoichiometry AOU / $\sum NO_x$. It is important to note that as we limit our analysis to waters well below the mixed layer, only water layers below 100 m can be investigated near the coast where the isopycnals are descending. Nevertheless the along-isopycnal spreading of subsurface waters between the offshore region and the seabed is well captured

by our local AOU approach. For simplicity and a more intuitive comparison with the OMP method, we multiply our results of the local AOU method with -1 and just refer to local oxygen anomalies.

## 2.7 Extended Optimum Multiparameter (OMP) method

A second approach, the OMP method, a water mass mixing analysis that also considers the bulk remineralization / respiration of nutrients and oxygen is following Karstensen and Tomczak (1998); Hupe and Karstensen (2000), is applied to separate remote and local processes. In brief, the extended OMP analysis decomposes observed conservative ($\Theta$, $S_A$) and non-conservative (nutrients, oxygen) parameters into water mass fractions of predefined source water types (see table 1 which also includes their weights). The decomposition is done by applying a non-negative least square fit in a multidimensional space (spanned by all parameters) following formula 4 in Karstensen and Tomczak (1998). For the conservative parameters only mixing fractions are resolved while for the non-conservative parameters mixing fractions and a bulk remineralization / respiration, controlled by a predefined set of stoichiometric ratios (here first guess Redfield ratio following Karstensen and Tomczak (1998), $8.625 = 138$ $O_2$ / $16 \sum NO_x$), is considered. The normalisation of the non-conservative parameters is described in Hupe and Karstensen (2000) in their section 3.1. As we are interested in the local remineralization (and respiration) we used source water types based on the data set at hand (see table 1) and considering potentially youngest NACW and SACW guided by maximum oxygen and minimum $\sum NO_x$ concentration within the nearby $\Theta(+/-0.04°C)/S_A+/-0.02g/kg)$ space (Fig. 2c, d). In this way the OMP method will provide us the local respiration and remineralization signals in the observational data.

Note the main differences between the AOU method and the OMP method is the reference frame: The AOU methods is based on the assumption that the regional AOU and $\sum NO_x$ fields are created from bulk respiration / remineralization of the reference profile. Mixing of water masses is not resolved. The OMP method decomposes the regional AOU and $\sum NO_x$ fields into mixing and a bulk remineralization / respiration, both in reference to pre-defined end members of water masses. We choose the most extreme SACW and NACW as our end members (Fig 2c, d) and thus only positive (or zero) bulk remineraliztion / respiration signal are obtained. In contrast, the AOU methods can provide positive and negative respiration / remineralization signals - simply because the reference profile does not represent an extreme profile of lowest oxygen / highest NOx or highest oxygen / lowest $\sum NO_x$. When comparing the results what matters most is that the range of the remineraliztion / respiration signal is similar. An advantage of the OMP method is that we can identify regions with high contribution of SACW or NACW and where regional remineralization / respiration should be assumed nearly unbiased by mixing. We apply both methods to test the robustness of the spatial distribution and temporal evolution of the estimated oxygen and $\sum NO_x$ anomaly patterns.

## 3 Results

### 3.1 Mean hydrographic structure along 18°N

Hydrographic observations along the 18°N were carried out between June 8 and 27, which represents the transition time between the upwelling season in winter and the low wind season in summer. During the cruise average alongshore winds of 10

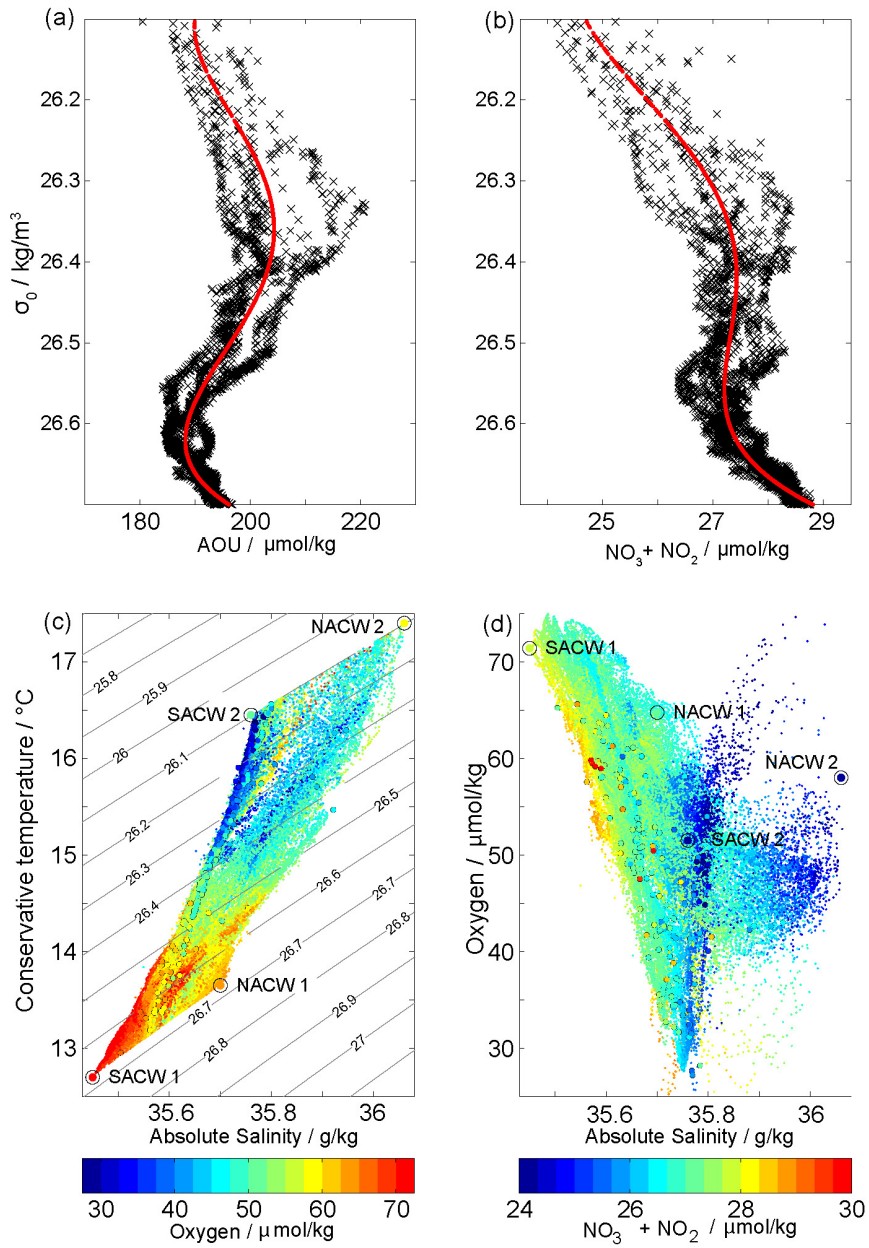

**Figure 2.** Apparent oxygen utilization (AOU) (a) and $\sum NO_x$ (b) reference profiles (black crosses) and polynomial fit (red) in density space. See location of offshore reference profiles in Fig. 1b. $\Theta/S_A$ (c) and $S_A$ / dissolved oxygen diagram (d) with dissolved oxygen and $(\sum NO_x)$ concentrations, respectively, in color shading. Glider-based SUNA and bottle data measurements are shown with small dots and dots with black edgings. Isopycnals are shown in black. The four water mass end members used for the OMP method are shown with extra large dots.

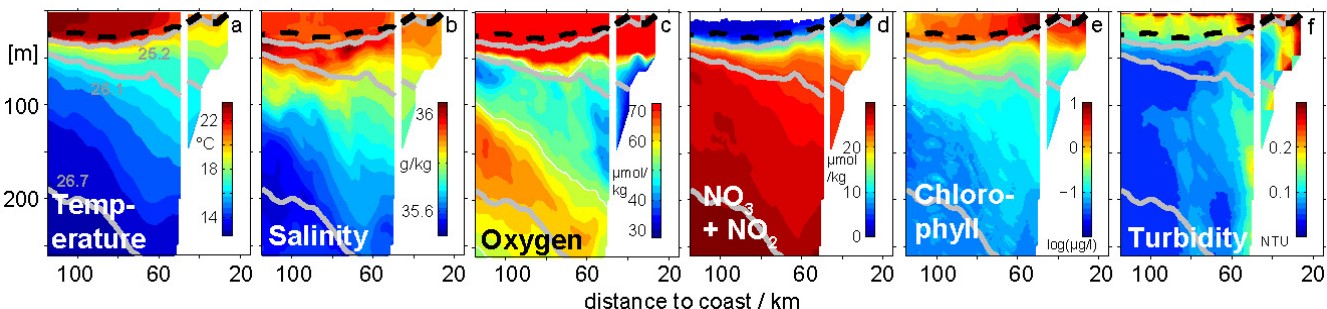

**Figure 3.** Mean cross-shore distribution of (a) conservative temperature ($\Theta$) in $^\circ$C, (b) absolute salinity ($S_A$) in g/kg, (c) dissolved oxygen ($O_2$) concentrations in $\mu mol/kg$ with oxygen concentrations of 50 $\mu mol\,kg^{-1}$ contoured in white, (d) nitrate + nitrite ($\sum NO_x$) concentrations in $\mu mol/kg$, (e) chlorophyll concentrations in $\mu g/l$ and (f) turbidity in NTU along 18$^\circ$N. The offshore (inshore) part separated by the white vertical line at 50 km is based on glider (CTD bottle) data. Averaging is performed in isopycnal space considering non outcropping isopycnals. Above the mean mixed layer depth averaging is done in depth space. Prior to the temporal averaging each of the 6 glider transects has been gridded on a regular 1 km grid via a simple 5 km rectangular window moving average. For consistency only CTD bottle data has been used for all parameters. The gray contours indicate $\sigma_\Theta$ 25.2, 26.1 and 26.7. The mixed layer depth (defined as the depth were the temperature equals the surface temperature minus 0.2$^\circ$) is shown in black dashed contour.

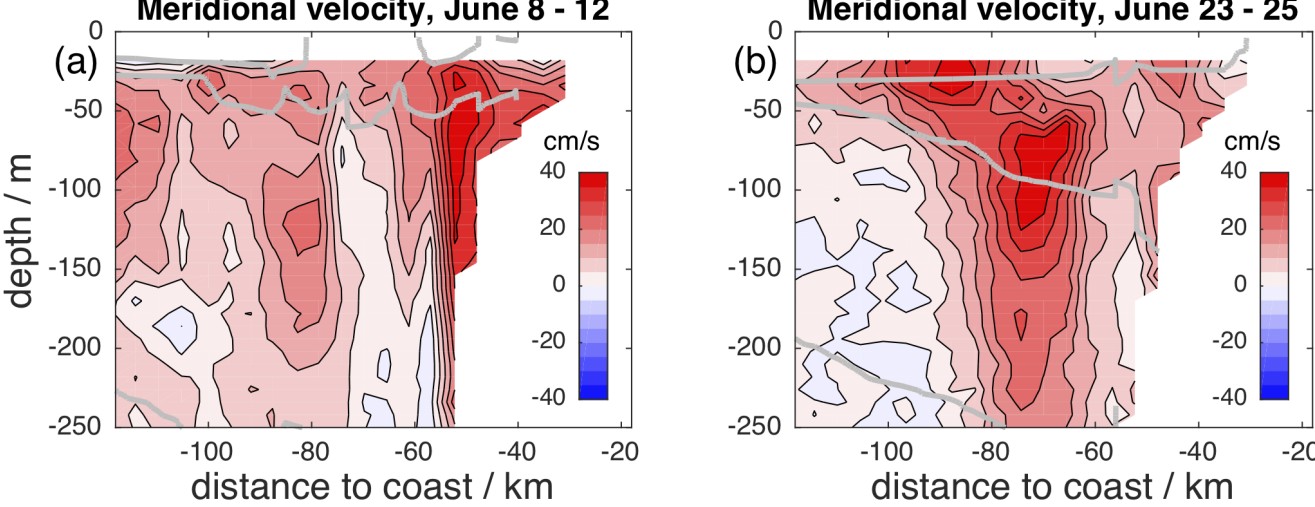

**Figure 4.** Meridional velocities in cm/s observed by vessel mounted acoustic Doppler current profiler along 18$^\circ$N between June 8 - 12 (a) and June 23 - 25 (b). Contour intervals are in 5 cm/s steps. Positive values represent northward flow. The isopycnals 25.2, 26.1 and 26.7 are contoured in gray.

**Table 1.** Source water mass type characteristics and weights for the OMP method described in section 2.7. The location of the source water mass types can also be seen in conservative temperature / absolute salinity and absolute salinity / oxygen space in Figure 2.

| Source Water Mass End Members | SACW1, $\sigma_\Theta = 26.7$ | NACW1, $\sigma_\Theta = 26.7$ | SACW2, $\sigma_\Theta = 26.1$ | NACW2, $\sigma_\Theta = 26.1$ | Weights |
|---|---|---|---|---|---|
| Conservative Temperature ($\Theta$ in °C) | 12.70 | 13.65 | 16.45 | 17.40 | 1 |
| Absolute Salinity ($S_A$ in g/kg) | 35.45 | 35.70 | 35.76 | 36.06 | 1 |
| Dissolved Oxygen ($O_2$ $\mu mol/kg$) | 71.40 | 64.75 | 51.50 | 58.00 | 0.5 |
| Nitrate + Nitrite ($\sum NO_x$ in $\mu mol/kg$) | 27.98 | 27.10 | 24.18 | 23.15 | 0.5 |
| Mass (in $kg$) | 1 | 1 | 1 | 1 | 10 |

m/s and 8.5 m/s were present at the offshore and inshore edge of the 18°N transect respectively (Fig. 1). As expected for this season we observe northward flow in the upper 250 m of the water column associated with the presence of the undercurrent (Klenz et al., 2018). The mean state of the circulation has been described in detail in Klenz et al. (2018) and thus we focus on the variability of meridional circulation in June 2014 (section 3.2).

The mean hydrographic cross-shore distribution presented in Figure 3 was constructed by combining ship- and glider-based measurements (see caption for details). As typical for a wind driven upwelling system, relatively cold surface water with temperatures down to 19.5°C were observed close (less than 25 km) to the coast (Fig. 3a). Further offshore the average surface temperature increased to maximum values of 23.7°C. The average mixed layer depth, here the depth were the surface temperature drops by 0.2 °C, is around 25 m offshore and reduces slightly to values around 20 m over the shelf break (Fig. 3).

In the offshore part of the section a rapid temperature drop is seen e.g. at 115 km offshore from 23°C in 25 m depth to 18°C in 40 m depth. Approaching the coast the vertical temperature gradient is less pronounced and below 40 m depth the slope of the isotherms even reverses i.e. the depth of the isopycnals decreases towards the coast. Furthermore the vertical spreading of the isopycnals results in a less stratified water body along the continental slope (Fig. 3a).

Highest absolute salinities of 36.08 g/kg were found offshore ($> 60$ km) and just below the mixed layer at 30 to 35 m depth

associated with the STUW (Fig. 3b). In the mixed layer itself slightly lower absolute salinity of around 35.96 g/kg is observed with minimal values of around 35.9 g/kg between 25 - 50 km offshore. The absolute salinity drops to minimal values of 35.55 g/kg at 200 m depth at 100 km and further offshore. Below the mixed layer on the shelf break between 40 and 150 m absolute salinities around 35.8 g/kg are observed near the sea floor.

In general highest oxygen concentrations of up to 240 $\mu\mathrm{mol\,kg}^{-1}$ are found within the mixed layer. Below 50 m depth the

oxygen concentrations drop rapidly to values well below 50 $\mu\mathrm{mol\,kg}^{-1}$ within the core of the shallow OMZ (Fig. 3c, white contour). Particularly low oxygen concentrations are found near the seabed with minimum mean oxygen concentrations down to 38 $\mu\mathrm{mol\,kg}^{-1}$ on the shelf-break at 150 m depth. The shallow OMZ reduces its vertical extent from about 100 m near the shelf-break to about 40 m further offshore. At 120 km from the coast the mean oxygen concentrations has again a local oxygen

maximum of 65 $\mu$mol kg$^{-1}$ at 150 m depth clearly separating shallow and deep OMZ. The depth of this intermediate oxygen maximum deepens towards the coast and is found at 250 m depth at 60 km from the coast. There mean oxygen concentrations of around 55 $\mu$mol kg$^{-1}$ are observed near the seabed and the separation between the shallow and deep OMZ is less pronounced.

Maximum $\sum NO_x$ concentrations of up to 30 $\mu$mol kg$^{-1}$ are found at 250 m depth 115 km offshore (Fig. 3d). In general $\sum NO_x$ decreases continuously towards the surface reaching values of around 10 $\mu$mol kg$^{-1}$ just below the mixed layer. While mixed layer $\sum NO_x$ concentrations $< 2$ $\mu$mol kg$^{-1}$ are found offshore ($> 60$ km to the coast), we observe an increase in mixed layer $\sum NO_x$ concentrations up to 10 $\mu$mol kg$^{-1}$ on the shelf indicating an upwelling signal. Maximum mean chlorophyll concentrations of up to 8 $\mu g/l$ are found in the mixed layer on the shelf break, located at about 50 km offshore. Further inshore the chlorophyll concentrations reach values between 4 to 7 $\mu g/l$. At the base of the mixed layer the observed concentrations range from 3.5 $\mu g/l$ at 70 km offshore down to 2 $\mu g/l$ further offshore. Below 50 m depth the chlorophyll concentrations values drop rapidly to values of well below 0.5 $\mu g/l$. Enhanced values of turbidity are observed in the surface layer with values increasing up to 0.3 NTU close to the coast. In contrast to chlorophyll we also observe higher turbidity (0.2 NTU) near the seabed at the shelf break (Fig. 3f). It is important to note that this higher turbidity signal is collocated with reduced oxygen concentrations (Fig. 3c). This collocation will be investigated in more detail in the single transect analysis (subsection 3.3.1). In summary we find in June 2014 a hydrographic structure along 18°N that shows typical upwelling pattern with cold and nutrient enriched waters at the surface observed over the shelf break. In accordance with the nutrient availability and enhanced chlorophyll concentrations our observations point to high primary productivity by phytoplankton during the observational period. One important observation is the occurrence of low oxygen concentrations close to the seabed where also enhanced turbidity is found.

## 3.2   Temporal change of the meridional circulation during June 2014

The along-shore circulation exhibited elevated variability during June 2014, which is described based on two ship-based acoustic Doppler current profiler transects (Fig. 3.2). During the beginning of the cruise between June 8 and 12 we observe a strong poleward flow of up to 40 cm/s on the shelf-break between 50 and 125 m depth (Fig. 3.2a). Further offshore in general weaker flow is found especially at greater depth. During the second period from June 23 to 25 2014 the undercurrent had moved offshore to about 70 to 100 km from the coast. The velocity maximum remains strong (about 40 cm/s) but was separated into two cores. One is found at a shallower depth above 50 m at 90 km offshore and the deeper in around 100 m depth at 70 km offshore. This displacement of the undercurrent results in much weaker velocities of around 10 cm/s at the shelf break (Fig. 3.2b). The change in the meridional velocity goes along with changes in the density field. While the 26.1 isopycnal is found at around 50 m depth across the whole transect during the first period, it deepens to 125 m depth at the shelf break about two weeks later. In section 3.3.2 we relate these circulation changes to the observed changes of the oxygen concentrations at the shelf break including additional glider-based depth averaged velocities.

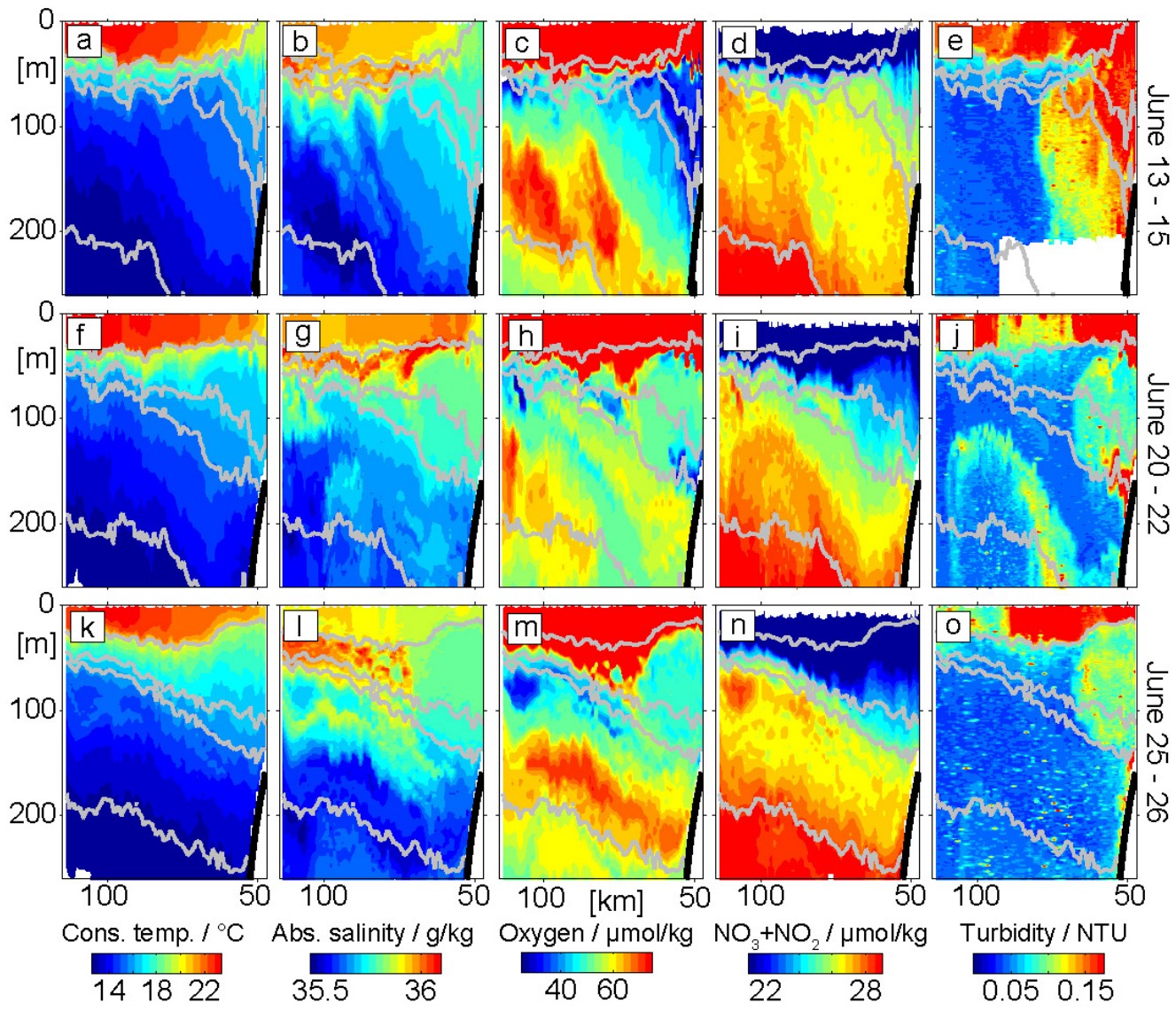

**Figure 5.** Snapshots of cross-shore distribution along 18°N of conservative temperature ($\Theta$) in °C (a,f,k), absolute salinity ($S_A$) in g/kg (b,g,l), dissolved oxygen ($O_2$) concentrations in $\mu mol/kg$ (c,h,m), nitrate + nitrite ($\sum NO_x$) concentrations in $\mu mol/kg$ (d,i,n), turbidity in NTU (e,j,o) for 3 different time periods June 13 - 15 (upper panel), June 20 - 22 (middle panel) and June 25 - 26 (lower panel). The isopycnals 25.2, 26.1, 26.28 and 26.7 are contoured in gray.

### 3.3 Oxygen and nitrate variability within the shallow oxygen minimum zone

In this section at first single glider transects will be used to investigate the synoptic oxygen and nitrate variability along $18°$N in June 2014 (section 3.3.1). Secondly the low frequency temporal change during the whole deployment will be investigated in section 3.3.2. The oxygen and nitrate variability will be described based on the observed bulk oxygen and nitrate distribution

(Fig. 5). Additionally we apply the two methods (AOU and OMP method) to the glider section data as introduced in section 2.6 and 2.7 respectively in detail. In short, with these methods we aim to investigate remote and local drivers of the observed variability (Fig. 6). The AOU method reveals local oxygen anomalies which either point to a local excess of oxygen (positive values) relative to the offshore reference profile or local oxygen loss (negative values) (Fig. 6b, g, l). The former are interpreted as a result of a remote ventilation while the latter are signals for local oxygen consumption. The OMP method also resolves

explicitly water mass composition and bulk oxygen and nitrate variability relative to the predefined source water masses. As we defined the source water masses to carry the maximum oxygen concentrations the OMP method only reveals negative oxygen anomalies pointing to local oxygen loss (Fig. 6d, i, n). For more details on the methods the reader is referred to section 2.6 and 2.7.

### 3.3.1 Synoptic oxygen and nitrate variability along single glider transects

The glider transects reveal high spatial and temporal variability in all observed parameters (Fig. 5, 6). Here we focus on three representative transects to investigate the observed oxygen and $\sum NO_x$ variability within the depth range of the shallow oxygen minimum zone (50 - 250 m). Our analysis reveals three different types of oxygen and $\sum NO_x$ anomalies: (i) positive oxygen anomalies associated with low salinity waters, (ii) negative oxygen anomalies associated with high turbidity signals especially near the seabed and (iii) low oxygen anomalies within non-turbid offshore waters. The oxygen anomalies always

show inverse relationships with $\sum NO_x$ indicating a close interplay between oxygen respiration and remineralization. In the following we discuss the three anomalies in more detail.

Patches of high oxygen concentrations are observed at depth between 100 m and 250 m at along the whole transects between June 13 and 26 (Fig. 5c, h, m). The size of these patches range from 20 to 50 km in the horizontal to 50 to 100 m in the vertical. Most of these high oxygen patches deepen downwards towards the coast and are aligned with the slope of isopycnals. A co-

location of high oxygen anomalies and low salinity patches is clearly visible e.g. along the first transect (June 13 - 15) at 100 km offshore (Fig. 5b, c). The OMP method shows that the low salinity is accompanied by high ($> 75\%$) SACW fractions (Fig. 5b, 6a) and point to the importance of SACW in supplying oxygen to the region. The local AOU method reveals that these high oxygen patches are associated with oxygen anomalies of the order of 10 $\mu$mol kg$^{-1}$ (Fig. 6b, g, l). The velocity observations point to a northward flow during June 2014. We conclude that these positive oxygen patches are caused by physical transport

of SACW into the region.

The second type of oxygen anomalies are low oxygen patches which exhibit a clear co-existence with enhanced turbidity signals. They are predominantly found near the seabed e.g. along the first two transects (June 13-15, June 20 - 22, Fig. 5c, h) where oxygen concentrations down to 29 $\mu$mol kg$^{-1}$ are observed (Fig. 5c). In transect 2 (June 20 - 22) also an elongated low

oxygen (45 $\mu$mol kg$^{-1}$) patch, aligned with enhanced turbidity, is found between 100 and 250 m depth and reaching more than 100 km offshore (Fig. 5h, j). The high turbidity signal may indicate offshore transport of resuspended organic matter which in turn may indicate locally enhanced microbial respiration near the sea floor. This hypothesis is further evaluated using a combination of UVP5-based particle abundance profiles (section 3.4) and CTD bottle-based DOM measurements (section 3.5). The oxygen concentrations associated with this anomaly seem to increase with time when comparing the section from mid June with the section from end of June (Fig. 5c, h, m) and goes along with a reduction of the turbidity signal. This low frequency changes become particularly visible in OMP method results (Fig. 6d, i, n) and will be discussed in more detail in section 3.3.2.

The third type of oxygen anomalies are low oxygen patches mainly found far offshore (about 100 km) and with horizontal and vertical scales of O(5 to 15 km) and O(20 to 40 m), respectively (Fig. 5c, h, m). One such example is seen during June 25 to 26 as a 10 km wide low oxygen (33 $\mu$mol kg$^{-1}$) and high $\sum NO_x$ (28.7 $\mu$mol kg$^{-1}$) lens about 110 km from the coast and in 80 to 100 m depth. The AOU method identifies this anomaly as a local oxygen and $\sum NO_x$ anomaly of + 10 $\mu$mol kg$^{-1}$ (Fig. 6i) and - 1.2 $\mu$mol kg$^{-1}$ (Fig. 6m) respectively within this feature. The OMP method reveals a higher respiration (about 25 $\mu$mol kg$^{-1}$) and $\sum NO_x$ remineralization (3 $\mu$mol kg$^{-1}$) signal (Fig. 6n, o) but with similar stoichiometry than the AOU method. The signal is associated with locally increased SACW factions (> 70%). But compared to the anomalies of the second type this anomaly shows very low oxygen concentrations pointing the a strong modulation of the initial water mass characteristics. No enhanced turbidity signal is associated with the anomaly.

In summary we identified 3 different types of oxygen and $\sum NO_x$ anomalies. The high oxygen anomalies at 150 to 250 m depth coincide with large fractions of low salinity SACW water. Two different low oxygen anomalies of presumably local formation are identified: One is mainly found at the shelf break close to the seabed and is associated with enhanced turbidity suggesting the signal may originate from the benthic boundary layer and was resuspended into the water column. The second type of low oxygen anomalies are found further offshore on similar isopycnal layers than the ones close to the shore. However, no enhanced turbidity signals are seen. This suggest that these offshore anomalies may originated from the shelf break but much longer ago and all turbidity signal is removed due to gravitational settling of particles. Both applied methods (AOU and OMP) gave similar oxygen respiration/nutrient remineralization patterns pointing to the robustness of these results. Our data does not allow to identify when the oxygen anomalies have been formed. However, we will use local particle-associated oxygen respiration rate estimates in section 3.4 to further evaluate the timescales that could explain the observed signals.

### 3.3.2   Temporal changes of the oxygen and nitrate during June 2014

Beside the high spatial variability of oxygen and nitrate along the single transects we also observe temporal variability during the 2 weeks of the glider observations. This temporal variability of oxygen and nitrate is particular pronounced near the seabed on the shelf break and will be investigated in more detail in the following.

The temporal change in the oxygen concentrations during our glider deployment is clearly visible also in the single transect perspective. While oxygen concentrations of 35 $\mu$mol kg$^{-1}$ were observed during the first transect (June 13 - 15) near the seabed on densities between 26.27 and 26.29 (Fig. 5c), these values increased to 50 $\mu$mol kg$^{-1}$ until the end of June (Fig. 5m).

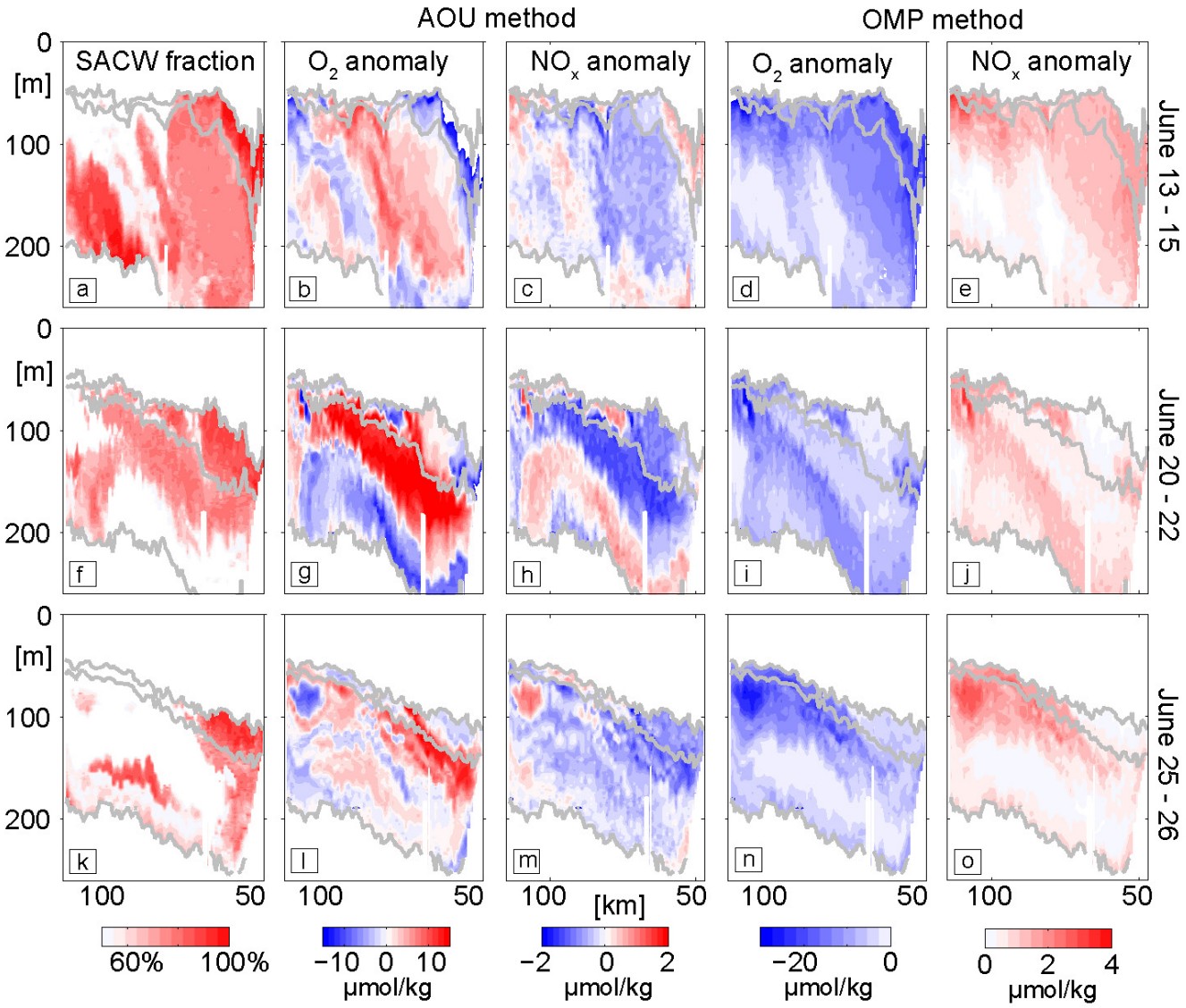

**Figure 6.** SACW fraction (a, f, k) in %, local oxygen (b, g, l) and nitrate + nitrite ($\sum NO_x$) (c, h, m) anomalies in $\mu mol/kg$ from AOU method (sect. 2.6) and local oxygen (d, i, n) and ($\sum NO_x$) anomalies (e, j, o) from OMP method (sect. 2.7) at 3 different time periods June 13 - 15 (upper panel), June 20 - 22 (middle panel) and June 25 - 26 (lower panel). The isopycnals 26.1, 26.28 and 26.7 are contoured in gray.

The low frequency temporal variability of $\sum NO_x$ is less clear due to its small magnitude of O(1-2 $\mu$mol kg$^{-1}$) in relation to the bulk $\sum NO_x$ concentrations of O(26.5 $\mu$mol kg$^{-1}$). In order to quantify this temporal variability and to investigate possible drivers we extract the oxygen and nitrate anomaly estimates from the AOU and the OMP method including the water mass fractions between the isopycnals of 26.27 and 26.29 close to the seabed (Fig. 7). Additionally, we analyse the temporal change of the meridional velocities on the shelf break. It can be seen that the observed changes in oxygen concentrations cannot be explained by variability in water mass composition. Changes in salinity are actually just around 0.015 g/kg (not shown). Thus the SACW fraction stays relatively constant near the seabed during the time of the observations (Fig. 7c) and even decreased slightly from 85% to 80%. Thus a slight decrease of SACW fraction should result in reduced oxygen concentrations if pure source water changes were responsible for the observed oxygen changes. However, the local AOU and OMP method reveal an increase in local oxygen anomalies of up to 15 $\mu$mol kg$^{-1}$ from June 13 to June 27. During the same time period the local $\sum NO_x$ anomalies reduce by about 1.2 $\mu$mol kg$^{-1}$.

During the glider deployment period we observe poleward velocities at the shelf-break. However with significant changes as already described in section 3.2. At the beginning of the glider deployment the undercurrent was found at the shelf break resulting in depth-average velocities of 30 cm/s between 47 and 55 km offshore (Fig. 7a). A strong increase in oxygen of about 10 $\mu$mol kg$^{-1}$ followed this strong velocity signal suggesting the advection of more oxygenated SACW (but similar temperature and salinity characteristics) from the south. During the rest of the deployment, the poleward velocity reduced to values between 0 and 10 cm/s. Despite the lower velocities it still seems that the poleward velocities were leading the changes in oxygen. This analysis suggests a strong modification of typical water mass signals, such as typically high concentrations of oxygen in SACW, are altered at the shelf due to locally enhanced respiration. Finally, although our observations suggest a relationship between increasing oxygen concentrations after periods of enhanced northward flow no increase in SACW fraction is observed pointing to a complex interplay between physical and biogeochemical processes.

In the following section we will investigate these high turbidity waters close to the seabed in more detail using UVP5, CTD sensor and bottle data to investigate the distribution and composition of the organic matter. In particular we aim to learn more about the importance and associated time scales of local organic matter remineralization processes and oxygen respiration rates.

## 3.4 Particle abundance and particle-associated oxygen respiration rate estimates

In this section we investigate the particle abundance along the transect using UVP5 observations to estimate particle-associated respiration rates. To relate particle abundance with oxygen respiration we use the lower bound parameterization for the Iversen et al. (2010) data following Kalvelage et al. (2015) (see section 2.2 for details). These respiration rate estimates are then used to investigate the possible role of local particle-associated oxygen consumption for the observed low oxygen patches described in section 3.3.1. Oxygen is continuously consumed within the interior ocean by microbial respiration processes but the magnitude of oxygen respiration varies both in time and space.

We distinguish here between two different particle size classes, where the diameter of small particles ranges from 0.14 to 0.53 mm (Fig. 8a) and for large particles from 0.53 to 16.88 mm (Fig. 8b). Particles of both size classes show qualitatively

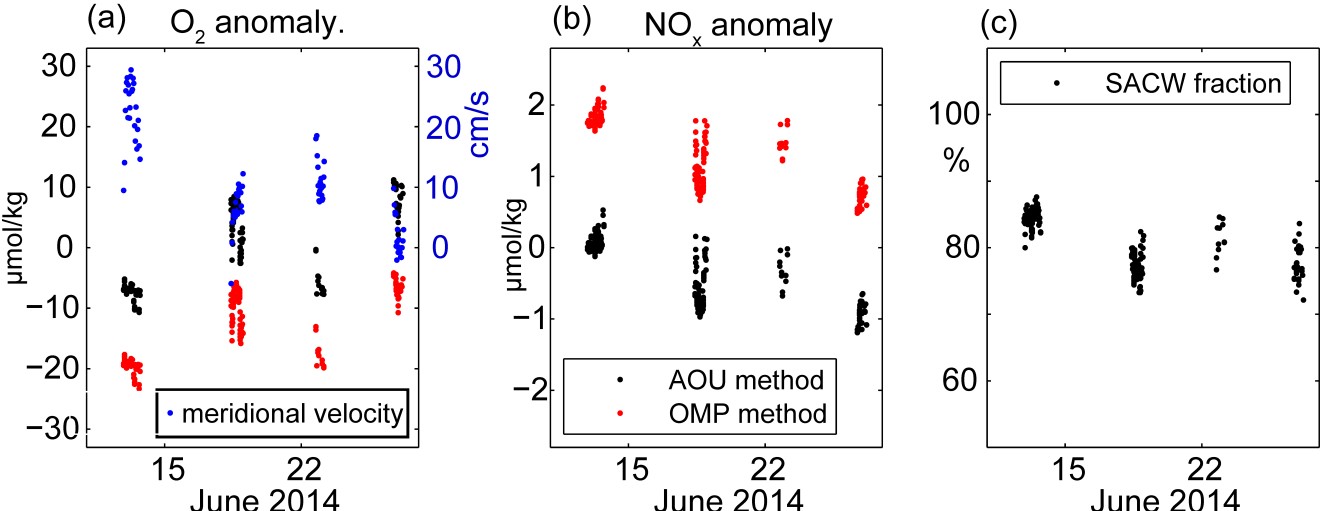

**Figure 7.** Temporal evolution of oxygen (a) and $\sum NO_x$ (b) anomalies (black: AOU method, red: OMP method) as well as SACW fraction (c) near the seabed (47 to 55 km from the coast) at densities between 26.27 and 26.29 in June 2014. The meridional glider-based depth-averaged velocity estimates are shown for all profiles between 47 and 55 km from the coast in panel a.

very similar distributions with enhanced particle abundance near the surface above the OMZ. On the shelf (50 m water depth) between 100 and 175 small particles and between 5 and 12 large particles $L^{-1}$ are observed. Additionally and more important for this study, there is another particle abundance maximum close to the seabed in the core of the OMZ at around 150 m depth. Here on average up 200 small and up to 10 large particles per liter are observed with the UVP5. The mean particle-associated

5 respiration rate estimates which are based on the combined abundance of small and large particles along 18°N during June 2014 reveal a similar pattern (Fig. 8c). Respiration rate estimates between 0.75 and 1.5 $\mu\mathrm{mol\,kg^{-1}\,day^{-1}}$ are found on the upper shelf at depths shallower than 50 m. At depth below 50 m enhanced respiration rate estimates are found near the seabed with a distinct maximum of 0.8 $\mu\mathrm{mol\,kg^{-1}\,day^{-1}}$ at 150 m.

Despite averaging/binning of repeated/close stations the particle abundance and the associated particle respiration rate esti-

10 mates along the transect remain patchy due to high variability. To better quantify its variability we separated all profiles into onshore (n = 49) and offshore (n = 12) profiles (not shown). Onshore (within 70 km from the coast) we estimated mean respiration rate of 0.57 +/- 0.44 $\mu\mathrm{mol\,kg^{-1}\,day^{-1}}$ in the upper 50 m of the water column and 0.34 +/- 0.26 $\mu\mathrm{mol\,kg^{-1}\,day^{-1}}$ at depth between 50 and 200 m. Offshore (more than 70 km from the coast) the rate estimates above 50 m water depth show mean values of 0.4 +/- 0.2 $\mu\mathrm{mol\,kg^{-1}\,day^{-1}}$. Below 50 m water depth they decrease to mean values of 0.11 +/- 0.08 $\mu\mathrm{mol\,kg^{-1}\,day^{-1}}$. In

15 summary we observe a typical standard deviation of the same order as the mean values. Importantly to note that variability is particularly elevated at around 50 m depth.

The oxygen respiration rate estimates can be used to learn about the approximate formation timescale of the observed low oxygen anomalies with high turbidity signals close to the seabed described in section 3.3.1. For this we combine the observed

magnitudes O(10 - 20 $\mu$mol kg$^{-1}$) of these anomalies with the estimated respiration rates and assume an alongshore advection of an enclosed water mass. This gives a formation time scale of about 12.5 to 25 days for oxygen anomalies of 10 to 20 $\mu$mol kg$^{-1}$ when using the rates near the seabed (0.8 $\mu$mol kg$^{-1}$ day$^{-1}$). Contrary the average interior offshore rates of about 0.1 $\mu$mol kg$^{-1}$ day$^{-1}$) result in formation timescales of about 100 to 200 days. Although we investigate the transect due to the availability of the data just in a two dimensional way we are aware about the fact that the observed water masses are advected through the transect. In the following we use a typical advection speed along the coast of about 0.1 m/s within the undercurrent close to the seabed (based on our velocity observations) and a constant alongshore respiration rates. This implies that the water masses move about 100 km during 12.5 days, which was the lower end of the formation time scale of these observed oxygen anomalies. Given the remote supply path of the SACW this is a relatively short distance and can still be referred to as local. Of course these are very crude first order estimates. As noted, we used the lower bound fit to the size-respiration rate data presented by Iversen et al. (2010) to obtain conservative rates. It seems reasonable to assume that the in situ respiration rates might be higher. Also the high variability of the respiration rate estimates both on- and offshore points to the possibility of short term high respiration events.

In summary, our results suggest that particle associated oxygen respiration close to the seabed is indeed able to change the oxygen concentrations within the study area during relatively short time periods. Thus beside advection of water mass properties via physical transport also fast local respiration might impact on the observed oxygen variability. In section 4 we further discuss the possible role of resuspension processes occurring within the bottom boundary layer.

In section 3.3.2 we describe an increase of oxygen concentrations close to the seabed despite relatively constant water mass properties. The transect has been sampled at relatively high temporal and spatial resolution including the repetition of various stations with the shipboard CTD/UVP5 measurements. Nevertheless there are not enough measurements available to make reliable estimates of temporal variability of respiration. However, the temporal change of the turbidity signal, i.e. a decrease of the turbidity signal near the seabed with time (Fig. 5e, j, o), suggests a reduction of particle-abundance and possibly the magnitude of particle-associated respiration during the study period. A reduction in respiration might be relevant for the low frequency temporal change in oxygen concentrations beside the observed changes in the circulation.

In summary high particle related respiration rates are estimated near the surface above the OMZ and at depth close to the seabed. Near the bottom we also find pronounced low oxygen patches in combination with high SACW fractions. As SACW brings oxygen into the region high oxygen concentrations might be expected. However the fact that we observe low oxygen patches points to a strong modification of typical water mass characteristics especially close to the coast driven by high local respiration.

## 3.5   Dissolved organic matter and amino acid distribution

As described in the previous section our UVP5 observations show enhanced particle abundance within the OMZ on the shelf break near the seabed. In this section we use CTD bottle-based measurements of DOC to investigate the potential role of DOC remineralization for the formation of the low oxygen anomalies close to the seabed as described in section 3.3.1. In particular

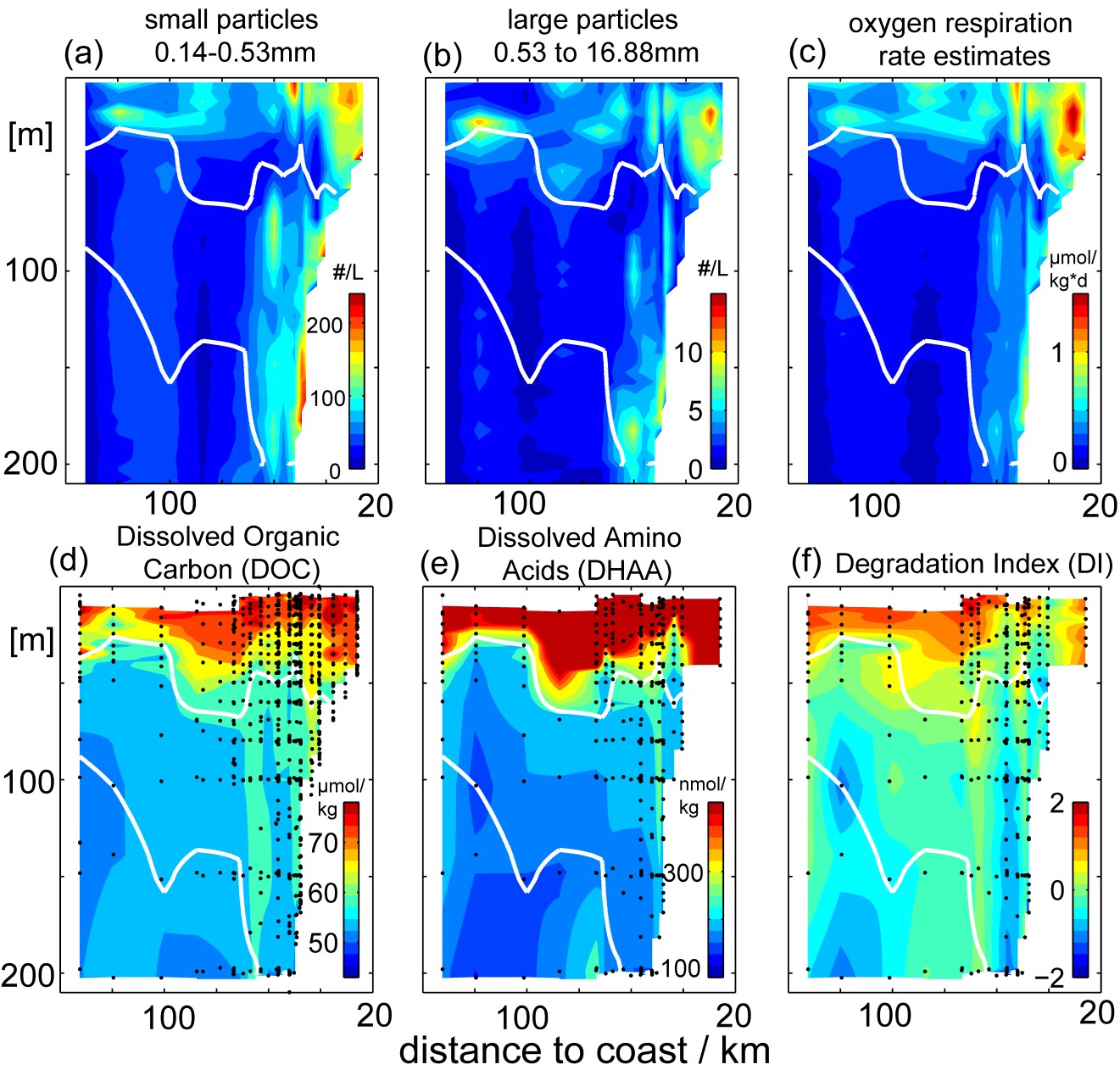

**Figure 8.** Small (a) and large (b) particle abundance, microbial particle-based oxygen respiration rate estimates (c), dissolved organic carbon (DOC) concentrations (d), dissolved amino acids (DHAA) (e) and DHAA based degradation index (DI) following Dauwe and Middelburg (1998) and Dauwe et al. (1999) (f) along $18°N$. Oxygen concentrations of $55$ $\mu\mathrm{mol\,kg}^{-1}$ are contoured in white. Most stations along the transect have been occupied multiple times and averaged in depth space prior visualization.

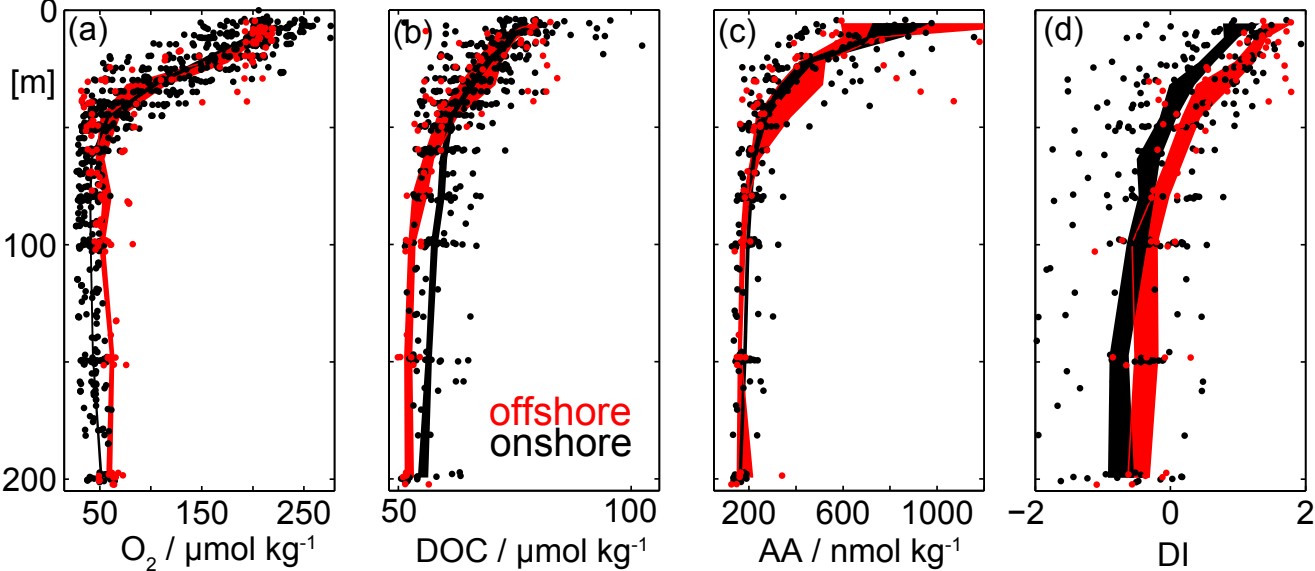

**Figure 9.** Offshore (red, > 70 km to the coast) and onshore (black, < 70 km to the coast) vertical distribution of dissolved oxygen ($O_2$) concentrations in $\mu mol/kg$ (a), dissolved organic carbon (DOC) in $\mu mol/kg$ (b), dissolved amino acids (DHAA) in $nmol/kg$ (c) and amino acid based degradation index (DI) following Dauwe and Middelburg (1998) and Dauwe et al. (1999) (d). Single measurements are shown as dots. Shaded area represents the mean +/- the standard error of the mean.

we want to address the question whether we also find higher DOC within the OMZ associated with the high turbidity and low oxygen signals. We furthermore quantify the DHAA concentrations and estimate a DHAA based degradation index (DI).

Highest DOC concentration of up to 100 $\mu mol/kg$ are found within the upper 50 m of the water column above the OMZ (Fig. 8d, 9b). DOC decreases rapidly to average values of around 60 $\mu mol/kg$ below 50 m within the oxycline (Fig. 9b). In the

5    upper 50 m of the water column no significant difference in the average DOC concentrations was determined between offshore (red, > 70 km to the coast) and onshore (red, < 70 km to the coast) stations (Fig. 9b). Below 50 m depth DOC concentrations of close to 70 $\mu mol/kg$ are found within the OMZ close to the coast, which are absent further offshore (Fig. 8d). Due to the patchiness of the DOC concentrations this becomes more clear when averaging all data points close to the coast and offshore data (Fig. 9b). Indeed this analysis reveals an increase of DOC of about 5 $\mu mol/kg$ within the OMZ close to the coast.

10    As seen for DOC also the highest DHAA concentrations of up to 1000 $nmol/kg$ are found in the surface layers (Fig. 8,9). DHAA concentrations decrease rapidly within the oxycline to average values of around 180 $nmol/kg$ at 100 m depths. The DHAA based DI shows positive values near the surface pointing to the production of fresh DOC. At depth mean DI values of well below 0 are observed. This suggests that the enhanced DOC at depth near the seabed likely results from particles dissolution and has already been reworked by microorganisms.

## 4 Discussion

In this study we investigate remote and local drivers of oxygen and nitrate variability off Mauritania in June 2014 using physical-biogeochemical glider-based observations in combination with ship-based profile and bottle data. Our data reveal different types of $O_2$ and $\sum NO_x$ anomalies driven by a combination of remote (primarily the transport of SACW in the boundary current system) and local (organic matter remineralization) processes. Our particle-related respiration rate estimates are enhanced in the upper oxycline close to the shelf break and near the seabed (Fig. 8c).

Brandt et al. (2015) found diapycnal oxygen flux divergences just below the mixed layer of similar magnitude as the estimated oxygen respiration of about 1 $\mu$mol kg$^{-1}$ day$^{-1}$. This suggests that the high oxygen consumption in the upper water column near the shelf off Mauritania might be largely balanced by diapycnal oxygen fluxes. Additionally SACW plays an important role for the oxygen supply into the region at the depth range of the shallow OMZ which has already been investigated in various studies (Peña-Izquierdo et al., 2012, 2015; Klenz et al., 2018). Our analysis support these findings as we find highest oxygen concentrations in combination with high SACW fractions. An early high-resolution modeling study by Glessmer et al. (2009) highlighted the role of the equatorial current system for the supply of water into the upwelling region off Mauritania. The good representation of this remote supply pathway in models might be crucial to capture remotely forced oxygen and nitrate variability off Mauritania. On contrary, Kounta et al. (2018) did not find a direct dynamical pathway between the North Equatorial Counter Current and the eastern boundary region of the ETNA. Nevertheless, poleward flow along the eastern boundary within the West African Poleward Boundary Current or the Mauritanian Current is well established but the flow exhibits elevated variability. Locally and remotely forced coastally trapped waves contribute to the variability (Klenz et al., 2018; Kounta et al., 2018). In fact, the observed offshore displacement of the undercurrent might be related to offshore Rossby waves progation that are shed by coastally trapped waves as described in Kounta et al. (2018). In their model, the authors find a decrease and even reversal (southward) of the poleward flow between 14.1N and 19.6N at 100 m depth between mid to end June (see their Fig. 12b). However to disentangle the exact drivers of the observed velocity changes within a highly turbulent environment under varying surface forcing is beyond the scope of this study.

Despite this important role of remote oxygen supply via SACW into the region we see also a strong decoupling between the physical water mass properties and the oxygen and nitrate concentrations due to enhanced local remineralization processes near the seabed. Our glider-based observations reveal negative $O_2$ anomalies of about 10 - 20 $\mu$mol kg$^{-1}$. This variability is huge when compared to the mean oxygen concentration of about 50 $\mu$mol kg$^{-1}$ within the shallow OMZ. These low $O_2$ anomalies cannot be explained by water mass variability but estimated local respiration rates are high enough to create these anomalies within time scales of a few weeks. These low $O_2$ anomalies are very pronounced close to the seabed were enhanced local particle-associated $O_2$ respiration associated with the resuspension of organic matter seems to occur. Iversen et al. (2010) observed particle size spectra further north of our study region near Cape Blanc. The authors also found a dominance of small particles and an increase of particle numbers with depth as also described by Nowald et al. (2006). This increase might be caused by lateral transport as suggested by modeling studies (Karakaş et al., 2006; Lovecchio et al., 2017). Here we focus

just on the upper 250 m of the water column but our observations also suggest the resuspension of organic matter and the consecutive offshore transport of this material.

Our main focus lies on the near shelf break area close to the seabed. However, we also find low $O_2$ / high $\sum NO_x$ anomalies further offshore which are located often just below the thermocline. These features do not carry high turbidity signals, which distinguishes them from the features closer to the coast. As we have no extensive oxygen respiration rate measurements in these features it is difficult to make final conclusions about their formation process. One possibility might be that they are simply older anomalies which have been formed close to the seabed but being transported offshore. This hypothesis is supported by the fact that they have the same densities as the low $O_2$ anomalies close to the seabed where enhanced respiration rates were observed. Another possibility might be local formation due to sporadically enhanced respiration rates. Previous studies in the area show that offshore respiration rates can vary significantly e.g. due to the presence of mesoscale eddies (Karstensen et al., 2015; Schütte et al., 2016) or upwelling filaments (Alvarez-Salgado et al., 2007). Also we find high variability in offshore particle abundance and the associated respiration rate estimates. So enhanced local respiration might also be responsible for the formation of these features. However, further research is needed to learn more about their exact formation processes. Regional high-resolution physical-biogeochemical model simulations which are calibrated with observations similar as those presented in this study might help to make progress on this topic.

Our study, which is based on a variety of approaches, points to the importance of enhanced local oxygen respiration for the observed oxygen and nitrate variability. The quantification of oxygen respiration however is associated with large uncertainties. Furthermore it is a major challenge to capture the spatial and temporal variability of oxygen respiration. Le Loeuff (1999) mentions anoxic conditions for benthic communities during severe climate conditions in the 1970 and 1980 further north on the Banc d'Arguin around 21°N. A bit further south of our study area at 14°N an anoxic event occurring on the continental shelf at water depth of around 30 m off Senegal was studied in detail by Machu et al. (2019). In order to understand the drivers of these events improved respiration rate estimates are needed. We thus want to stress the importance of further research to reduce the errors of these estimates. This would allow an improved calibration of regional model simulations to possibly simulate or even predict these anoxic events.

Additionally to the enhanced particle abundance we also find enhanced DOC concentrations close to the shelf (Fig. 8,9). So far the DOC distribution has not been described for this region. However, further north in the Canary upwelling system DOC was identified to play a major role in mesopelagic respiration (Santana-Falcón et al., 2017). Our observations suggest that DOC plays also an important role for the remineralization process within the study area. Our analysis of the quality of the DOC revealed that the DOC at depths is mainly formed due to resuspension of organic matter and the dissolution of particles during microbial decomposition near the seabed. The focus in this study lies on the organic matter dynamics close to the seabed. But we also observe high concentrations of DOC at the outer edge of the transect pointing to the importance of offshore DOC transport as already suggested by Alvarez-Salgado et al. (2007) for the northern part of the upwelling area.

Beside pelagic oxygen respiration also benthic oxygen uptake is driven by the export of organic matter from the surface. Dale et al. (2014) quantified the oxygen flux into the sediment off Mauritania in 2011 and estimated $10\, mmol/m^2/d$ in the depth range between 50 and 100 m and a decrease to $3\, mmol/m^2/d$ at greater depth. These rates seem to play a minor role

for the oxygen budget of the deep OMZ (Brandt et al., 2015). This is thought to be due to the large volume of the deep OMZ and the relatively small benthic uptake rates at depth below 300 m. (Dale et al., 2014; Brandt et al., 2015). However, locally benthic uptake rates might be important for the oxygen distribution especially at shallower depth (less than 44 km offshore) where higher benthic uptake rates are found and the water column is less deep (Brandt et al., 2015). It is important to note,

that Brandt et al. (2015) compared the benthic oxygen uptake rates on the upper shelf with the pelagic oxygen respiration rate estimates for the deep OMZ O(1 - 5 $\mu$mol kg$^{-1}$ year$^{-1}$). However, our particle-based oxygen respiration estimates presented in this study and the diapycnal oxygen flux divergences estimates in Brandt et al. (2015) are of the order of 1 $\mu$mol kg$^{-1}$ day$^{-1}$ . Thus when comparing the benthic rates of oxygen uptake from Dale et al. (2014) to the determined pelagic respiration rates in a layer of 50 m vertical extend above the sediments, benthic oxygen consumption is an order of magnitude smaller than our

pelagic respiration estimates. Dale et al. (2014) further estimated that the sediment is a net sink for $\sum NO_x$ with fluxes of about 1.5 $mmol/m^2/d$ (see their table 2). Thus if benthic oxygen and nitrate uptake would dominate over pelagic remineralization proccesses we would expect to see negative $\sum NO_x$ anomalies in conjunction with the low oxygen anomalies near the seabed. However, we see positive $\sum NO_x$ anomalies which points to the dominance of pelagic remineralization processes.

## 5   Conclusions

A combination of interdisciplinary high-resolution glider and ship based observations are used to investigate the remote and local drivers of oxygen and nitrate variability in the shallow oxygen minimum zone off Mauritania. This analysis includes the role of water mass characteristics and local organic matter remineralisation processes. We observe high oxygen variability of the order of 20 $\mu$mol kg$^{-1}$ which represents about 40 % of the bulk mean oxygen concentrations of 50 $\mu$mol kg$^{-1}$ . Some part of this variability can be explained by water mass composition namely the fraction of SACW which is an important oxygen

source for the region. However this clear water mass characteristic is modified strongly by local oxygen respiration in particular closer to the coast near the seabed. There local oxygen respiration is able to create oxygen anomalies of the same magnitude as the water mass variability within time scales of 1 - 2 weeks. This finding points to the importance of local organic matter remineralisation for the oxygen distribution and variability off Mauritania (Fig. 10). The observed ($\sum NO_x$) variability of 1-2 $\mu$mol l$^{-1}$ only represents about 5 % of the bulk nitrate concentrations. Thus we conclude that local remineralisation processes

are particularly important for the oxygen distribution and variability off Mauritania. The Mauritanian upwelling regime, as a typical eastern boundary upwelling system (EBUS), is important for the global carbon cycle but also of local relevance due to ecosystem services and fish catch. How the oxygen distribution and its variability will change in the future is thus of high relevance. However even state-of-the-art high-resolution coupled physical-biogeochemical model simulations of the Mauritanian OMZ fail to reproduce the vertial structure of the OMZ off Mauritania i.e. the intermediate oxygen maximum is

missing (Duteil et al., 2014). This might be related to a weak representation of the remote ventilation with SACW. Furthermore, our observations point to complex local biogeochemical processes e.g. the resuspension and remineralization of organic matter at the seabed, which is not captured by simple biogeochemical models (Duteil et al., 2014). As both remote oxygen supply mechanisms and local remineralization processes are expected to change in a future climate, multi-scale model simulations

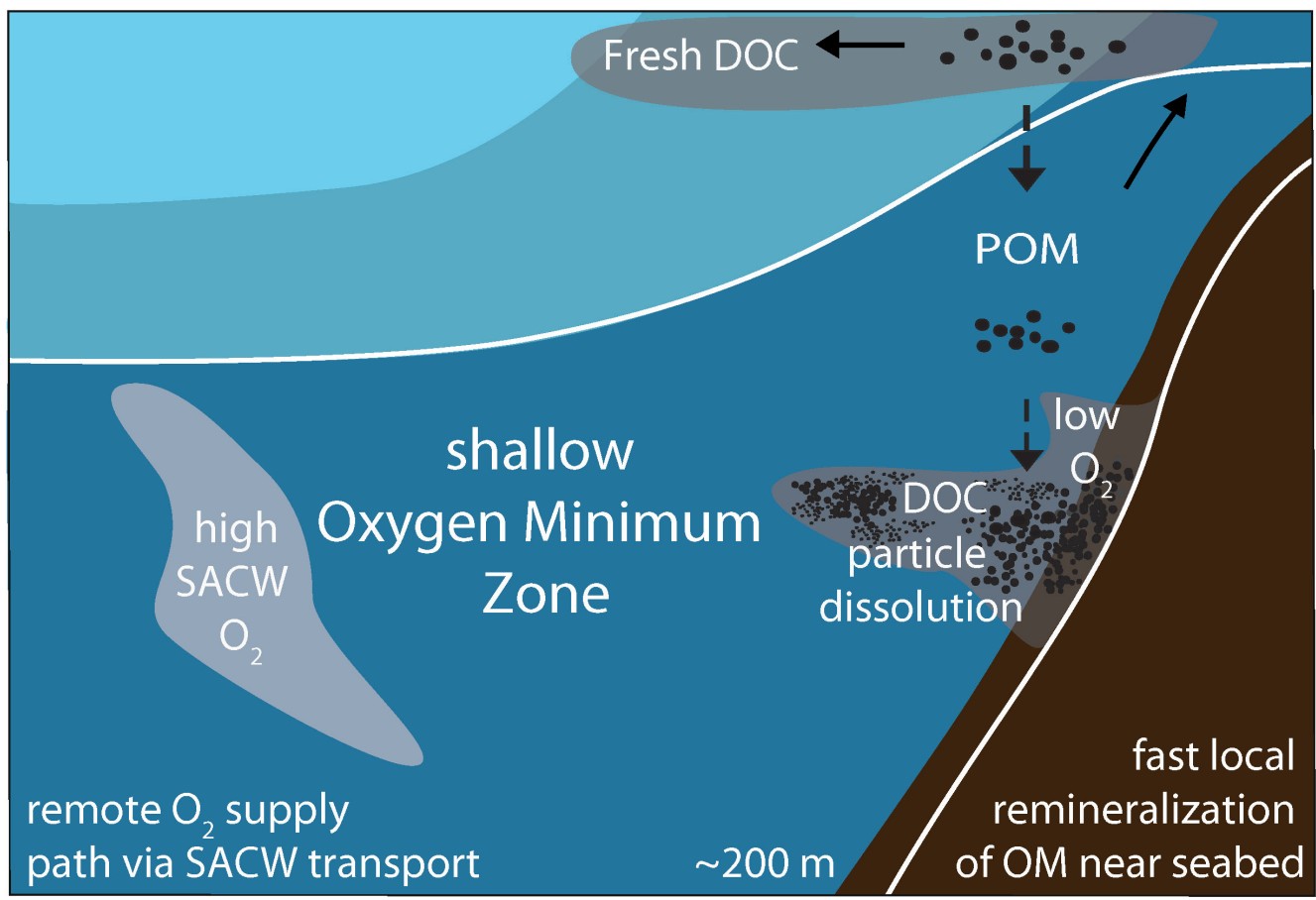

**Figure 10.** Schematic showing the interplay between remote physical transport mechanisms and local organic matter remineralization processes creating oxygen and nitrate variability within the shallow oxygen minimum zone (approximately within the upper 200 m of the water column) off Mauritania in June 2014. Offshore a strong coupling between water masses signals (e.g. high SACW fractions) and enhanced oxygen concentration is observed (light grey shading). Close the the seabed fast local organic matter remineralization processes occur in combination with particle dissolution and the release of DOC. The associated respiration results in the formation of low oxygen anomalies (dark gray shading).

are needed which capture large scale changes as well as fine-scale physical and biogeochemical processes near the coast. Our observations are valuable to evaluate and possibly improve these simulations to give more reliable projections on the future evolution of oxygen and nitrate distribution and variability off Mauritania and linked changes in primary productivity, plankton abundance and finally fishery yields.

*Code and data availability.*  The CTD sensor and nutrient bottle data can be downloaded freely at https://doi.pangaea.de/10.1594/PANGAEA.860480 and https://doi.pangaea.de/10.1594/PANGAEA.885109 respectively. The glider data can be accessed at https://doi.pangaea.de/10.1594/PANGAEA.877388. The Underwater Vision Profiler (UVP) data can be accessed at https://doi.pangaea.de/10.1594/PANGAEA.885759. The organic matter dataset can be accessed at https://doi.pangaea.de/10.1594/PANGAEA.896321. The vessel mounted ADCP dataset can be accessed at https://doi.pangaea.de. According to the SFB 754 data policy (https://www.sfb754.de/de/data), all data associated with this publication will be published at a world

data center (www.pangaea.de, search projects:sfb754) when the paper is accepted and published. Upon publication all code necessary for the data analysis and preparation of the figures of this manuscript will be freely available on https://zenodo.org/ under the following doi: 10.5281/zenodo.1689927.

*Author contributions.*  ST designed and carried out the experiment on board of Meteor in cooperation with MD. ST carried out the data analysis and wrote the main manuscript. JK motivated the water mass analyses methods applied in this study and contributed to the writing

of the manuscript. GK carried out the CTD and glider sensor data processing and calibration including the novel glider-based $(\sum NO_x)$ measurements. RK calculated the UVP5-based particle-related oxygen respiration rate estimates and assisted with the interpretation of the UVP5 dataset. MD contributed to the interpretation of the velocity measurements. AE contributed to the interpretation of the organic matter datasets and wrote parts of the manuscript. All co-authors reviewed the manuscript and contributed to the scientific interpretation and discussion.

*Competing interests.*  The authors don't see any competing interests in this work.

*Acknowledgements.*  Special thanks goes to the captain and the crew of R/V Meteor for their support during the M107 cruise. We further thank the authorities of Mauritania for the permission to work in their territorial waters. We thank B. Domeyer, A. Bleyer, U. Lomnitz, R. Suhrberg, S. Trinkler and V. Thoenissen for the nutrient analyses. We thank R. Flerus for the water sampling on board. We thank R. Flerus and J. Roa for the dissolved organic matter analyses and C. Begler for preparing and carrying out the glider deployment and the

navigation. This study is a contribution of the Sonderforschungsbereich 754 "Climate - Biogeochemistry Interactions in the Tropical Ocean," sub-projects B6, B8 and B9 which is supported by the German Research Foundation. S. Thomsen further receive funding be the European Commission (Horizon 2020, MSCA-IF-2016, WACO 749699: Fine-scale Physics, Biogeochemistry and Climate Change in the West African Coastal Ocean).

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
