# Peer review of "Remote and local drivers of oxygen and nitrate variability in the shallow oxygen minimum zone off Mauritania in June 2014"

_Biogeosciences, 2018_

## Referee Comment (RC1) · Anonymous Referee #1 · 24 Aug 2018

This manuscript describes a series of glider and ship-borne hydrographic and biogeo-chemical measurements in the oxygen minimum zone (OMZ) off Mauritania. Through a series of cross-shelf glider transects between 13 and 26 June 2014, the authors show highly variable oxygen distributions within and around the shallow oxygen minimum. Using analytical tools (OMP, etc.) and an outstanding set of biogeochemical data, they identify the role of local (remineralization of organic matter) vs. distant (penetration of the relative oxygen-rich SACW) drivers of the oxygen distribution. The manuscript is well written, the data set is extensive, the analytical methods are clever and useful, and the results are interesting and relevant for the dynamics of the OMZ. I can definitely support its publication but some concerns have to be addressed first. In my opinion,

the main weakness of the manuscript is the lack of dynamical insights. From my point of view, the paper would be much improved by including a description of the circulation patterns in the area and a discussion of their relation to the oxygen distributions. Other than that, I have only some suggestions for the improvement of some particular points and minor comments.

GENERAL COMMENTS: - As reported in previous studies, the authors claim that the ventilation of the upper thermocline waters is mainly driven by the "remote" supply of SACW with the boundary currents along the African shelf. The dynamics of these currents are described in the introduction, and also mentioned in the discussion in relation to their representation in ocean models. However, in the manuscript, the remote ventilation of the OMZ by SACW is only studied with an "static" interpretation of the water masses distribution. I miss a more dynamical characterization of the study area at the time of sampling. Could you show some ADCP velocities to illustrate what is the mean circulation during the glider samplings and the impacts on O2 distribution? In figures 3 and 4, strong isopycnal tilting is observed at the shelf break, suggesting some meridional flow. Does this relate to the observed high-oxygen anomalies associated with the penetration SACW or with the location of the OMZ?

- I also have some concern regarding the description of the oxygen anomalies in relation to the proportion of SACW. From Figure 4 it seems that the relationship between relatively high-oxygen concentration and the presented distribution of SACW is by far not univocal. For example, the lowest oxygen concentrations close to the shelf break between the 26.1 and 26.28 isopycnals coincide with high proportions of SACW. The TS diagram (Fig 2) and the distributions of the different variables (Figs. 3-5) seem to suggest that the ventilation of the study area is carried out mostly by the densest variety of SACW. I would suggest to separate the contribution of both SACW end-members in figure 4a,f,k and this would help to illustrate how the densest variety ventilates the area (if my interpretation is right). This is something similar as Peña-Izquierdo et al. (2012) did by defining local and a remote varieties of SACW.

- The authors describe a third type of oxygen anomaly (low oxygen with non-turbid offshore waters), but they do not describe their drivers in much detail. These anomalies are rather persistent and have some interesting characteristics: they are located just below the pycnocline, they are observed only with the OMP analysis, and generally associated with low contribution of SACW, but there is sometimes an enhancement of SACW in their core. These anomalies correspond generally to relatively high salinities, and I could associate them with the low oxygen values located along the line (15°C, 35.7 g/kg) – (16°C, 35.9 g/kg) in the TS diagram. With this information and some hint about the circulation in the area, the authors could speculate a bit more about the origins of this water body.

SPECIFIC COMMENTS:

1 – Section 3.1. Could the authors show the reference profiles for the AOU analysis and the polynomial fit in the T/S or O2/S diagram? I think it could help to the interpretation of some O2 anomalies and the differences with the OMP method.

2 – Section 3.2. I think a more detailed description of the OMP method would be desirable. I would like to see which equations have actually been solved. For example, how do you exactly implement the resolution of non-conservative tracers, do you solve the NO (NO = O2 + r*NO3, where N is the Redfield O2:NO3 ratio) (quasi-)conservative tracer? Also, what is the weight given to each equation?

3. Figure 4. Can you label the end-members in this figure? They are difficult to identify, particularly in panel B. The oxygen color scale in this figure is inverted with respect to Figures 3 and 4.

4 – Section 3.2, Figure 4 and corresponding description. The oxygen and nitrate anomalies obtained with the AOU and OMP methods have different definitions of the zero value, which makes a bit difficult to identify differences between both methods. You could solve that by choosing different predefined O2 values for the end-members used to compute the anomaly. Instead of using the extreme values (highest oxygen

concentration around the end-member) you could use some measure of the mean O2 value around each end-member. This would make the OMP anomalies more evenly distributed around zero.

5- Section 4.1. I would suggest to briefly outline in the text how the distributions in Figure 3 were generated. You could also show the distribution of mixed layer depth in Figure 3, as it is extensively described in the text.

TECHNICAL CORRECTIONS:

P4 – L24: "low frequent" → "low frequency"

P6 – L17: "See also (Yücel et al., 2015) and (Fiedler et al., 2016) [...]" → "See also Yücel et al., (2015) and Fiedler et al., (2016) [...]". There are a number of other references for which the parenthesis are not properly placed, particularly in the discussion section. Please revise them: P20 – L9, P21 – L2, P21- L13.

P11 – L7: Add a reference to Figure 3b at the end of this sentence: "Highest absolute salinities of 36.08 g/kg were found offshore (> 60 km) and just below the mixed layer at 30 to 35 m depth associated with the STUW."

P11 – L19: Add a reference to Figure 3d at the end of this sentence: "NOx concentrations towards the surface of up to 30 $\mu$mol kg$-1$ are found at 250 m 115 km offshore"

P14 – L11: "The OMP method reveals a a higher respiration [...]", please remove one "a"

P16 – P5: " The continues* advection of SACW from the south via the boundary current system is an oxygen source for the density levels discussed here", continues→continuous

P17 – L12: "[. . .] at 150 m depths", remove depths

P21 – L11/14: "Nevertheless it is important to note that we observe high concentrations of DOC at the outer edge of the transect pointing the importance of offshore DOC

transport as already suggested by (Alvarez-Salgado, 2007) for the northern part of the upwelling area": important → importance, (Alvarez-Salgado, 2007) → Alvarez-Salgado, et al.. (2007). This reference is incorrect in the reference list, there are multiple coauthors of this paper.

——————————————————

---

## Referee Comment (RC2) · Anonymous Referee #2 · 15 Oct 2018

"Remote and local drivers of oxygen and nitrate variability in the shallow oxygen minimum zone off Mauritania in June 2014" by Soren Thompsen et al.

The authors present an analysis of the variability of dissolved oxygen (DO) and nitrate (NOx) in an upwelling region off Mauritania (18°S) using a series of glider and ship-based observations including a novel underwater vision-profiler mounted on the rosette to obtain particle size spectra in the water column. In addition to the CTD, the glider was equipped with oxygen and nitrate (Satlantic Deep SUNA) sensors. Based on water-mass analysis and AOU changes (along with a sequence of reasonable hypothesis) the authors separate DO and NOx local variability from a remote signal. The remote

signal is mainly associated to changes in ocean transport, while local processes are related to local respiration and remineralization. Particularly, the results based on glider observations showed that an increasing of turbidity is related to negative DO anomalies close to the bottom and the authors hypothesize that resuspended particles increase local respiration. This hypothesis is supported by observations from the vision-profiler and, DOC and DHAA data obtained from the ship.

In general, the paper is well structured and the data processing is well done and adequately explained (references are appropriate when additional information is required). Figures and figure texts are clear and main features are well emphasized. I have only few comments:

1.- I'm not sure if I missed anything, but it's not clear to me how the particle-based oxygen respiration rates were calculated. In the Page 17 lines 8 and 9, the sentence: "the mean particle associated respiration rate estimate along 18 $^\circ$ N during June 2014 reveal a similar pattern". On what is this claim based? The spatial distribution of the particles has been described in the previous lines, but the respiration rates are not mentioned. How the values of respiration rate given below (values in lines 10 to 12) are then obtained. Are these values given by OMP / AOU methods (like in Figure 6)?

From the sentence indicated above (lines 8 and 9) should it be understood that both sets of particles (small and large) are associated with a same respiration rate?

2.- Regarding the above comment, I understand that values for respiration (and nutrient remineralization) rate are crude estimations, nevertheless it would be valuable if the authors can provide an estimation of the error (or the variability, based on how they were estimated) associated to the given numbers.

3.- I wonder why there is no mention to the third type of oxygen anomalies described in section 4.2.1 in the discussion. I mean the negative (positive) DO (nitrate) anomaly "lens about 110 km from the coast and in 80 to 100 m depth" showed in 4n, m (and also in figure 5, last row).

Minor specific comments

Some extra information about the glider data would contribute to improve the paper. Did de glider sample only until 250 m depth? What is the accuracy and precision of the nitrate data? Does the G2 glider have a pumped CTD? Was the sensor a fast-response Aanderaa optode?

Page 7. L 27: AOU was already defined (in P4 L31)

P 8. L 20-21: Check redaction.

P 13. L 11: OPM should be OMP

P 13. L 25: delete "by"

P 17. L 23: "Of cause" should be "Of course"

P18. L16: "waver" should be "water"

P 23. L 9: Define OM or you mean DOM

Table 1: include units

Figure 3: Indicate what the white vertical line represents.

Figure 7. Delete one "following".

Figure 9. A depth scale for the study region would help (because the shallow OMZ was studied).

―――――――――――――――――

---

## Author Comment (AC1) · 27 Nov 2018

**This manuscript describes a series of glider and ship-borne hydrographic and biogeochemical measurements in the oxygen minimum zone (OMZ) off Mauritania. Through a series of cross-shelf glider transects between 13 and 26 June 2014, the authors show highly variable oxygen distributions within and around the shallow oxygen minimum. Using analytical tools (OMP, etc.) and an outstanding set of biogeochemical data, they identify the role of local (remineralization of organic matter) vs. distant (penetration of the relative oxygen-rich SACW) drivers of the oxygen distribution. The manuscript is well written, the data set is extensive, the analytical methods are clever and useful, and the results are interesting and relevant for the dynamics of the OMZ. I can definitely support its publication but some concerns have to be addressed first.**

> We thank Reviewer #1 for the constructive in depth review which helped us significantly to improve the manuscript.

**In my opinion, the main weakness of the manuscript is the lack of dynamical insights. From my point of view, the paper would be much improved by including a description of the circulation patterns in the area and a discussion of their relation to the oxygen distributions. Other than that, I have only some suggestions for the improvement of some particular points and minor comments.**

> We agree with the main point of reviewer #1 that the first version of the paper was lacking on dynamical insights. Thus we now include a description of the meridional circulation observed during the measurement program and relate the boundary current variability to the oxygen distribution.

**GENERAL COMMENTS: - As reported in previous studies, the authors claim that the ventilation of the upper thermocline waters is mainly driven by the "remote" supply of SACW with the boundary currents along the African shelf. The dynamics of these currents are described in the introduction, and also mentioned in the discussion in relation to their representation in ocean models. However, in the manuscript, the remote ventilation of the OMZ by SACW is only studied with an "static" interpretation of the water masses distribution. I miss a more dynamical characterization of the study area at the time of sampling. Could you show some ADCP velocities to illustrate what is the mean circulation during the glider samplings and the impacts on O2 distribution? In figures 3 and 4, strong isopycnal tilting is observed at the shelf break, suggesting some meridional flow. Does this relate to the observed high-oxygen anomalies associated with the penetration SACW or with the location of the OMZ?**

> Please note that we decided at first to leave out the dynamical description as it is partly covered in Klenz et al. (2018) (now published). However, as Klenz et al. (2018) focusses more on seasonal timescales and just shows one mean velocity transect, we decided to include the description of the alongshore circulation focusing on its temporal variability during June 2014. For this we use both ship-based ADCP measurements and glider-based velocity estimates. Throughout the cruise we find positive meridional (northward) velocities (as described in Klenz et al. 2018) but with significant short-term variability.

Glider-based depth-averaged velocity estimates are added to Fig. 7a to illustrate variability of meridional velocity and its relation to the observed changes in oxygen. Especially at the beginning of the glider deployment we observe strong northward flow of up to 30 cm/s which is followed by a drastic increase of oxygen concentrations with a time delay of a few days. During the remainder of the deployment the velocities on the shelf break reduce significantly to values between 10 cm/s and 0 cm/s. The ship-based ADCP data reveals that an offshore displacement and shallowing of the undercurrent is responsible for the observed reduction of the meridional velocities on the shelf-break. We added a figure of two ADCP sections from two different time intervals of the alongshore and cross-shore velocity to show this properly. We conclude that the increase in oxygen seen during our observations must result from advection of oxygen from the south. Interestingly this does not go along with a change in water mass properties or an increase of SACW fraction. We only see a very small change in salinity and even a slightly decrease of SACW. We think that it is important to stress that a strong decoupling between classical physical water mass properties and biogeochemical parameters occurs and thus interdisciplinary approaches are needed. We stress this is in the abstract too.

However, we also have to accept that although we have included velocities now we still have a 2D view which is very limiting and does not allow to explain the small scale synoptic tracer distributions e.g. single patches of high SACW or low oxygen etc based on our observations. As we have learned during previous 3D glider-swarm experiments that the 2D approach is not sufficient (please also look into the discussion in Thomsen et al. 2016). One main reason is that without a 3D sampling its not possible to know the initial water mass properties which are advected through a transect within a very short time. Here we assumed that in general higher oxygenation waters are found in the south. Thus it seems reasonable for us to assume that high velocities will result in a slightly delayed increase in oxygen. We believe that the additional velocity data and the resulting new discussion improved the manuscript.

**- I also have some concern regarding the description of the oxygen anomalies in relation to the proportion of SACW. From Figure 4 it seems that the relationship between relatively high-oxygen concentration and the presented distribution of SACW is by far not univocal. For example, the lowest oxygen concentrations close to the shelf break between the 26.1 and 26.28 isopycnals coincide with high proportions of SACW. The TS diagram (Fig 2) and the distributions of the different variables (Figs. 3-5) seem to suggest that the ventilation of the study area is carried out mostly by the densest variety of SACW. I would suggest to separate the contribution of both SACW end-members in figure 4a,f,k and this would help to illustrate how the densest variety ventilates the area (if my interpretation is right). This is something similar as Peña-Izquierdo et al. (2012) did by defining local and a remote varieties of SACW.**

Indeed high proportion of SACW is not always associated with highest oxygen concentrations (both in spatial and temporal evolution) this is because the SACW can stay for a while in the region and respiration will alter the oxygen (and likewise nutrients) in a proportion that is unrelated to the mixing of conservative tracers. We already discussed this in the initial manuscript (e.g at the end of section 4.2.2 old manuscript). We extended this discussion (now section 3.2.2) and state for example: "This analysis suggests a strong modification of typical water mass signals, such as typically high concentrations of oxygen in SACW, are altered at the shelf due to locally enhanced respiration." We conclude in the paper that classical water mass analysis suffers from not resolving changes in oxygen and nutrients though biogeochemical cycling and eventually resolves this by modifying water mass fractions. Thus we perform this interdisciplinary study to show that respiration rates are highly variable in space. The study by Peña-Izquierdo et al. (2012) differs from our

study in two main points: (1) they do not consider the biogeochemical cycling of nutrients and (2) they introduce a virtual water mass (regional SACWcv) which makes interpreting the field a bit complex. In fact (1) and (2) are related. SACWcv was introduced by Peña-Izquierdo et al. (2012) because "the most common water in the shadow zone is the regional SACWcv variety, which is substantially saltier than SACW. Furthermore, the SACWcv has the lowest oxygen concentrations of the three central water masses (Fig. 3b), suggesting that the SACWcv water mass is an old SACW variety that has lost most of its oxygen content while also mixing slowly with the nearby salty NACW"

[Figure]

Figure 1: Temporal change in oxygen in µmol/kg (left), conservative temperature in °C (middle) and absolute salinity in g/kg near the seabed (47 to 55km from the coast) at densities between 26.27 and 26.29 in June 2014.

So this additionally defined SACWcv is characterized by slightly higher salinity but lower oxygen concentrations due to long residence time. However, we observe a significant increase of oxygen concentrations (+15 µmol/kg) near the coast but almost no change in salinity characteristics. We attached a figure below to illustrate this here. Although the changes in salinity are not relevant in our view (just an increase of 0.015 g/kg between the first and second near coastal observations) they even contradict the expectations when following the SACW and SACWcv definitions. We see an increase in salinity but an increase of oxygen. Furthermore we want to point to the fact that the oxygen concentrations decrease again significantly towards June 22 but in this case the temperature and salinity stays literally constant. We already discussed parts of this in the original paper (now section 3.2.2, before 4.2.2, "temporal changes of oxygen and nitrate in June 2014"). This water furthermore comes from the south as revealed by our additional velocity analysis. Given that we don't see drastic changes in pure physical water mass characteristics but large changes of oxygen content we don't think that a more complex water mass framework will be useful to explain this variations in oxygen. Please note that Peña-Izquierdo et al. (2012) limited their analysis to data below 100-150 m depth in order to avoid the influence of biogeochemical processes.

On contrary as we think that a multi-parameter analysis - including physical processes and biogeochemical cycling is actually needed and this is what we carried out here. The result from this analysis suggests that very high local respiration rates close to the seabed are able to alter water mass properties (e.g. here nitrate and oxygen concentrations) in time

periods of a few weeks. Indeed the lighter water masses are more impacted by this local processes due to the hydrographic situation (slope of isopycnal = descending to coast, and in general higher rates closer to the surface). Thus they also show a large variety of oxygen concentrations (Review Figure 1). We added some more details on this e.g. we mention now also the absolute change in salinity which is very small.

A new study by Klenz et al. (2018) which focusses on the seasonal variability of the boundary current system and the hydrography also highlights the strong seasonal modification of oxygen characteristics in relation to water mass characteristics (SACW) near the shelf break at 18°N and discuss possible local physical and biogeochemical drivers. Their Figure 9 shows the mean SACW fraction and oxygen concentrations from two cruises in March 2011 and in June 2014 (same cruise as presented in this paper but they just use mean properties). Although they observe a higher fraction of SACW during the June 2014 cruise compared to the March 2011 cruise, they observe actually lower oxygen concentrations in June 2014. We conclude that more research focussing on seasonality is needed to really understand these changes which are not explainable by pure physical water mass analysis.

**- The authors describe a third type of oxygen anomaly (low oxygen with non-turbid offshore waters), but they do not describe their drivers in much detail. These anomalies are rather persistent and have some interesting characteristics: they are located just below the pycnocline, they are observed only with the OMP analysis, and generally associated with low contribution of SACW, but there is sometimes an enhancement of SACW in their core. These anomalies correspond generally to relatively high salinities, and I could associate them with the low oxygen values located along the line (15°C, 35.7 g/kg) – (16°C, 35.9 g/kg) in the TS diagram. With this information and some hint about the circulation in the area, the authors could speculate a bit more about the origins of this water body.**

We agree with the reviewer that the third type of oxygen anomalies could receive a bit more attention in the discussion (also suggested by Reviewer 2). In general its difficult to say much about the exact "drivers" and origin of these anomalies as our measurements (especially the sparse in-situ particle abundance measurements) themselves are insufficient to fully understand their origin and formation. Although without 3D high-frequent sampling as mentioned above we learned from previous studies that studying such small features is not possible.

The observed offshore respiration rates cannot explain their creation in a very short time period. However, we might have missed some short term extreme events e.g. associated with strong local respiration events (e.g. due to localised high primary production and the associated export of organic matter e.g. driven by eddies (Karstensen et al. 2015, Schuette et al. 2016) or filamentary structures further offshore (Alvarez-Salgado et al. 2007). As the review mentioned correctly, these anomalies are found just below the pycnocline, this might suggest that local respiration associated with near surface processes might drive these anomalies. Our in-situ observations (particle abundance and respirations rates) show high variability and thus might allow such short term extreme events but are not sufficient to fully answer this question. Another explanation might be that they have been formed near the seabed (same density) but all the particles have already settled due to gravitation or have been remineralised. In this case they would just be an old type of the low O2 anomalies described in detail.

Please note that both methods reveal these anomalies but with different magnitude due to conceptional differences in both methods. We added a summary of the discussion above to the discussion section (now section 4) of the revised version of the manuscript (first version section 5, discussion).

**SPECIFIC COMMENTS:**
**1 – Section 3.1. Could the authors show the reference profiles for the AOU analysis and the polynomial fit in the T/S or O2/S diagram? I think it could help to the interpretation of some O2 anomalies and the differences with the OMP method.**

We thank the review for this suggestion. We tried to come up with a meaningful incorporation of the 18 reference profiles for the AOU analysis as well as the polynomial fit into the T/S or O2/S space. However, concluded in the end that this might lead to even more confusion. Instead we added now a plot of the reference profiles (AOU and NOx) in density space to Fig. 2. By this we aim to make more clear now that the AOU method does not incorporates any T/S relationships. Its purely based on the density vs. AOU/NOx relationship.

[Figure]

Figure 2: Absolute salinity / conservative temperature (a) and absolute Salinity / dissolved oxygen diagram (b) with dissolved oxygen and NOx concentrations, respectively, in color shading of the offshore reference profiles used for AOU method. The water mass end members of the OMP methods are show for comparison.

Indeed each of the 18 reference profiles for the AOU analysis has an own T/S/O2/NOx relationship with a certain variability especially at shallower depths (see review Fig. 2). For the AOU approach we used a mean AOU or NOx profile in density space obtained by a polynomial fit (see new Fig. 2a, b). This polynomial fit cannot be presented in T/S space as it has no specific temperature or salinity characteristic (but density). Of course we could create also an average temperature and salinity profile and add this to the TS plot. However, to include this and the single reference profile into the T/S space would somehow suggest that the AOU method incorporates T/S signals which is not the case.

However, to allow the reviewer a detailed comparison as requested we added a figure to the review. It shows the 18 reference profiles in T/S and S/O2 space including the end members of the OMP method. The colorbar range is as in Fig. 2 to allow a comparison. Given that the readers can access the full review after publication we hope that this is a satisfactory solution.

Please note that its not the aim of the paper to make a detailed methodical comparison between both methods. We rather wanted to test the robustness of the spatial distribution and the associated temporal variability of the O2 and NOx anomalies.

**2 – Section 3.2. I think a more detailed description of the OMP method would be desirable. I would like to see which equations have actually been solved. For example, how do you exactly implement the resolution of non-conservative tracers, do you solve the NO (NO = O2 + r*NO3, where N is the Redfield O2:NO3 ratio) (quasi-)conservative tracer? Also, what is the weight given to each equation?**

Thank you for this comment. We used here the standard set of linear equations that link the mixing of conservative parameters (unknowns are for this part the water mass fractions) with an additional terms that solve for the bulk remineralization/respiration signal (for the non-conservative parameters). The bulk remineralization/respiration is introduced through a first guess Redfield ratio (see Karstensen and Tomczak 1998). This technique and the equations are not new and have been used many times and thus we feel it is sufficient to cite the former work that introduced the technique.

The reviewer referred to what is also known as the "quasi-conservative" approach (that goes back to work by Broecker & Peng 1982 in "Tracers in the Sea"). The "quasi-conservative" approach is different as such that it does not allow to resolve the remineralized/respired contribution (bulk remineralization) but substitute it by creating a linear combination of two properties (in the reviewers example it was oxygen and nitrate). In our extended OMP method the bulk remineralization is explicitly solved - which is only possible by adding it into the mixing equations as a correction term.

We added more details to the reference Karstensen and Tomczak (1998) (equation 4). In Hupe and Karstensen (2000) section 3.1. introduces the normalisation of the non-conservative parameters. Also the weights for to each parameter are given in table 1 now - the weights are empirically estimated.

**3. Figure 4. Can you label the end-members in this figure? They are difficult to identify, particularly in panel B.**

We labeled the end-members in both panels.

**The oxygen color scale in this figure is inverted with respect to Figures 3 and 4.**

We changed the color scale in figure 3 and 4 to be more consistent with figure 2. All have the same range now and non is inverted.

**4 – Section 3.2, Figure 4 and corresponding description. The oxygen and nitrate anomalies obtained with the AOU and OMP methods have different definitions of the zero value, which makes a bit difficult to identify differences between both methods. You could solve that by choosing different predefined O2 values for the end-members used to compute the anomaly. Instead of using the extreme values (highest oxygen concentration around the end-member) you could use some measure of the mean O2 value around each end-member. This would make the OMP anomalies more evenly distributed around zero.**

We thank the reviewer for this useful comment. Indeed both methods have different definitions of the zero value due to their different conceptual approach. The AOU approach assumes, as described in section 3.1/2.6 (old manuscript/ revised manuscript), that the offshore water (reference profile) spreads towards the coast along isopycnals. This is a simple 2D view of an upwelling region often found in the literature. However, the fact that the AOU method creates positive anomalies points to the importance of along-shore advection (confirmed by the now included circulation data) as obviously the reference offshore profile did not capture the highest oxygen anomalies associated with SACW.

On contrary the OMP method separates mixing of water mass fractions and bulk remineralization in respect to the defined source water types. Indeed we choose the youngest and thus most extreme SACW and NACW as our end members (revised manuscript Fig 2c, d) and thus only positive (or zero) bulk remineraliztion / respiration signal are obtained. As the parameter space (T, S, O2, NOx) has to be covered consistently (youngest NACW and SACW) it is actually not possible to make both methods more similar by changing the oxygen values of the end member to something like "mean values" as suggested by the reviewer.

However, in our opinion the important result is that both methods give the same range (between maximum and minimum values, see colorbar ranges), similar spatial pattern and a robust temporal evolution despite quite different approaches. We also think its valuable that the AOU method reveals positive signals related to water mass transport. Thus we decided to keep both methods as performed in the initial version of the manuscript. However, we extended the discussion on the differences between the two methods at the end of section 2.7 (revised manuscript).

**5- Section 4.1. I would suggest to briefly outline in the text how the distributions in Figure 3 were generated. You could also show the distribution of mixed layer depth in Figure 3, as it is extensively described in the text.**

Thanks for pointing out that more clarity is need how the distributions in Fig. 3 were generated. Please note that a detailed description of the procedure was already given in the caption of Figure 3 itself. As this text is quite technical and we wanted to keep the focus of the text in section 3.1 (revised manuscript) on the scientific results we decided to keep this detail to the caption. As this is probably a matter of taste we hope that the reviewer can accept our decision. However, please note that we now further improved the figure caption (mention the white space, which represents the transition between glider (offshore) and CTD (onshore) data and further made a reference to the figure caption in the result parts of section 3.1. (revised manuscript) at the beginning.

**TECHNICAL CORRECTIONS:**
**P4 – L24: "low frequent" → "low frequency"**

Changed. (the reviewer was referring to P4. L. 34 (old manuscript)

**P6 – L17: "See also (Yücel et al., 2015) and (Fiedler et al., 2016) [...]" → "See also Yücel et al., (2015) and Fiedler et al., (2016) [...]". There are a number of other references for which the parenthesis are not properly placed, particularly in the discussion section. Please revise them: P20 – L9, P21 – L2, P21- L13.**
Corrected.

**P11 – L7: Add a reference to Figure 3b at the end of this sentence: "Highest absolute salinities of 36.08 g/kg were found offshore (> 60 km) and just below the mixed layer at 30 to 35 m depth associated with the STUW."**

We thank the reviewer for improving the manuscript with this useful suggestions which will help the reader to take up the message quicker. Reference added.

**P11 – L19: Add a reference to Figure 3d at the end of this sentence: "NOx concentrations towards the surface of up to 30 µmol kg–1 are found at 250 m 115 km offshore"**
Reference added.

**P14 – L11: "The OMP method reveals a a higher respiration [...]", please remove one "a"**

Thanks! One "a" was removed.

**P16 – P5: " The continues* advection of SACW from the south via the boundary current system is an oxygen source for the density levels discussed here", continues→continuous**
Corrected.

**P17 – L12: "[. . .] at 150 m depths", remove depths**
Removed

**P21 – L11/14: "Nevertheless it is important to note that we observe high concentrations of DOC at the outer edge of the transect pointing the importance of offshore DOC transport as already suggested by (Alvarez-Salgado, 2007) for the northern part of the upwelling area": important → importance, (Alvarez-Salgado, 2007) → Alvarez- Salgado, et al.. (2007). This reference is incorrect in the reference list, there are multiple coauthors of this paper.**

Sentence modified and corrected. Citation corrected within the whole paper.
Please note that when downloading the official *.bibtex file from the Wiley page there is only one author. https://aslopubs.onlinelibrary.wiley.com/doi/10.4319/lo.2007.52.3.1287?cookieSet=1

**Literature:**

Alvarez-Salgado, X. A., Arístegui, 5 J., Barton, E. D., and Hansell, D. A.: Contribution of upwelling filaments to offshore carbon export in the subtropical Northeast Atlantic Ocean, Limnology and Oceanography, 52, 1287–1292, 2007.

Karstensen, J., Schütte, F., Pietri, A., Krahmann, G., Fiedler, B., Grundle, D., Hauss, H., Körtzinger, A., Löscher, C. R., Testor, P., Vieira, N., and Visbeck, M.: Upwelling and isolation in oxygen-depleted anticyclonic modewater eddies and implications for nitrate cycling, Biogeosciences, 14, 2167–2181, 2017.

Karstensen, J. and Tomczak, M.: Age determination of mixed water masses using CFC and oxygen data, Journal of Geophysical Research: Oceans, 103, 18 599–18 609, 1998.

Klenz, T., Dengler, M., and Brandt, P.: Seasonal Variability of the Mauritania Current and Hydrography at 18°N, Journal of Geophysical Research: Oceans, 0, 2018

Hupe, A. and Karstensen, J.: Redfield stoichiometry in Arabian Sea subsurface waters, Global Biogeochemical Cycles, 14, 357–372, 2000.

Peña-Izquierdo, J., Pelegrí, J. L., Pastor, M. V., Castellanos, P., Emelianov, M., Gasser, M., Salvador, J., and Vázquez-Domínguez, E.: The continental slope current system between Cape Verde and the Canary Islands, Scientia Marina, 76, 65–78, 2012.

Schütte, F., Karstensen, J., Krahmann, G., Hauss, H., Fiedler, B., Brandt, P., Visbeck, M., and Körtzinger, A.:    Characterization of "deadzone" eddies in the eastern tropical North Atlantic, Biogeosciences, 13, 5865–5881, 2016.

Broecker & Peng, in Tracers in the sea, Lamont-Doherty Geological Observatory, Columbia University, 1982 - 690, pages

---

## Author Comment (AC2) · 27 Nov 2018

**The authors present an analysis of the variability of dissolved oxygen (DO) and nitrate (NOx) in an upwelling region off Mauritania (18°S) using a series of glider and ship-based observations including a novel underwater vision-profiler mounted on the rosette to obtain particle size spectra in the water column. In addition to the CTD, the glider was equipped with oxygen and nitrate (Satlantic Deep SUNA) sensors. Based on water- mass analysis and AOU changes (along with a sequence of reasonable hypothesis) the authors separate DO and NOx local variability from a remote signal. The remote signal is mainly associated to changes in ocean transport, while local processes are related to local respiration and remineralization. Particularly, the results based on glider observations showed that an increasing of turbidity is related to negative DO anomalies close to the bottom and the authors hypothesize that resuspended particles increase local respiration. This hypothesis is supported by observations from the vision-profiler and, DOC and DHAA data obtained from the ship. In general, the paper is well structured and the data processing is well done and adequately explained (references are appropriate when additional information is required). Figures and figure texts are clear and main features are well emphasized. I have only few comments:**

> We thank the reviewer for his/her motivating and supporting review.

**1.- I'm not sure if I missed anything, but it's not clear to me how the particle-based oxygen respiration rates were calculated. In the Page 17 lines 8 and 9, the sentence: "the mean particle associated respiration rate estimate along 18 ° N during June 2014 reveal a similar pattern". On what is this claim based? The spatial distribution of the particles has been described in the previous lines, but the respiration rates are not mentioned. How the values of respiration rate given below (values in lines 10 to 12) are then obtained. Are these values given by OMP / AOU methods (like in Figure 6)?**
**From the sentence indicated above (lines 8 and 9) should it be understood that both sets of particles (small and large) are associated with a same respiration rate?**

> We thank the reviewer for this comment which reveals that the text needs more clarity about the way the respiration rates were estimated. Please note that in section 2.2 we give detailed information about this and also the relevant references. However, we changed the sentence accordingly to make more clear that the respiration rates are based on the combined abundance of small and large particles. Additionally we now give reference to the method description detailed in section 2.2. Furthermore we now refer to Fig. 8c at the end of the sentence to make clear that we are not referring to the AOU or OMP method estimates here.

> We further decided to merge the initial "Observational datasets and data processing" section with the "method" section as we realised that several methods (e.g. the description of the particle-associated respiration estimation was found in the "Observational datasets and data processing" section. We hope that this is now clearer for the reader. Please note that this led to a change in the labelling and numbering of the sections as we have one section less now.

**2.- Regarding the above comment, I understand that values for respiration (and nutrient remineralization) rate are crude estimations, nevertheless it would be valuable if the authors can provide an estimation of the error (or the variability, based on how they were estimated) associated to the given numbers.**

We fully agree with the reviewer that it is very important to discuss and mention the limitations of our respiration rate estimates. We now address the main uncertainties of the particle-associated oxygen respiration rate in a quantitative manner as asked by the reviewer. For this we reached out to M. Iversen to receive the original dataset which is used in Kalvelage et al. (2015) to construct the relationship between particle size and oxygen respiration rates. The main uncertainty results from the large variability of respiration rate measurement of individual particles. This variability probably reflects the high variability of particle associated respiration which depends on the energy level, the quality and the state of the particles as well as the status of colonisation by microbes. Additionally it seems to be particularly difficult to analyse very small particles experimentally in the lab. However small particles make up a large amount of the overall particle abundance and thus are potentially also very important for the oxygen respiration rates. We decided to follow the lower-bound approach by Kalvelage et al. (2015) as they give most reasonable results when comparing with various other estimations (e.g. Brandt et al. 2015, diapycnal flux divergence off Mauritania along 18°N). However, to stress the uncertainty we added the information that the PARR could also be up to a factor of 13 to 16 higher, based on the upper bound fit to the individual particle respiration rates as shown in Iversen et al. 2010, Fig. 8c. Despite this large uncertainty we want to note that for our study its already very useful to quantify the relative difference between the onshore (near-bottom) and offshore particle abundance and the associated rate estimates.

Additionally to the in depth error discussion above we give now a quantification of the variability of the calculated particle-associated respiration rate estimates by separating all profiles into onshore and offshore profiles. This is necessary to have enough profiles to estimate a proper standard deviation. Although most stations have been occupied actually several times there are a few stations mainly offshore which were just occupied once. This makes it difficult to give an estimate on the variability there when describing the overall distributions across the whole transect. However, we added ranges of observed values for the specific regions of interest. In general the variability is high and standard deviations of the order of the observed mean values are typical.

We hope that this additional in-depth discussion will result in more future efforts to reduce this uncertainty. Thus we now also stress this in the discussion in relation to a recent study of Machu et al. (2018) which reports for the first time of an anoxic event on the continental shelf off Senegal. Improved local respiration estimates are needed to calibrate regional model simulations to possibly predict these kind of events.

We added the following sentence to the figure caption to make it more clear to the reader how the transect has been constructed. "Most stations along the transect have been occupied multiple times and averaged in depth space prior visualization." We further changed to a non-continuous colorbar in Fig. 8 (reduced numbers of colors) to make it easier to read the values.

**3.- I wonder why there is no mention to the third type of oxygen anomalies described in section 4.2.1 in the discussion. I mean the negative (positive) DO (nitrate) anomaly "lens about 110 km from the coast and in 80 to 100 m depth" showed in 4n, m (and also in figure 5, last row).**

We added a paragraph of discussion on the third type of DO anomalies into the discussion section. Please also see our answer to a similar comment of Reviewer 1 above.

**Minor specific comments**
**Some extra information about the glider data would contribute to improve the paper.**

We agree with the reviewer that some more detailed information regarding the glider data and how the glider was navigated would improve the paper.

**Did de glider sample only until 250 m depth?**

    The glider was programmed to dive and sample only until 300 m depth as we focus on the upper OMZ. We added a sentence to section 2.5.

**What is the accuracy and precision of the nitrate data?**

    The accuracy and precision of the Suna nitrate sensor is given with 2 µmol/l and 0.3 µmol/l in the data sheet. https://www.seabird.com/asset-get.download.jsa?id=54627862138
Our in-situ recalibration using temporally and spatially close measurement of glider-based nitrate measurements and in-situ CTD bottle based data reveal an even better accuracy (RMS) of 1.3 µmol/l. When reducing the fit to values at depth below 50 m (the focus of this study) we even get a better fit due to smaller internal variability. We added this information including these values to section 2.5.

**Does the G2 glider have a pumped CTD?**

    Yes the glider has a pumped CTD. This information was added to section 2.5.

**Was the sensor a fast-response Aanderaa optode?**

    The glider was equipped with a normal Aanderaa optode with a time constant of about 20 - 30s. However, our data processing includes beside the normal calibration (Hahn et al. 2014) a correction of this time delay as described in Bittig et al. (2014). Me mention that we use Aanderaa optodes now. However, as all details of the data processing and calibration are described in very detail in Thomsen et al. (2016), they are not repeated again in this manuscript.

**Page 7. L 27: AOU was already defined (in P4 L31)**

    Changed

**P 8. L 20-21: Check redaction.**

    Removed.

**P 13. L 11: OPM should be OMP**

    Corrected.

**P 13. L 25: delete "by"**

    Deleted.

**P 17. L 23: "Of cause" should be "Of course"**

    Changed.

**P18. L16: "waver" should be "water"**

    Changed.

**P23. L9: Define OM or you mean DOM**

    Thanks for checking the manuscript so carefully! This is very valuable. We changed OM to organic matter. We refer here to both POM and DOM and thus organic matter is used. This was also changed in the figure caption of the final schematic.

**Table 1: include units**

    Units have been included.

**Figure 3: Indicate what the white vertical line represents.**

    The vertical line represent the transition between the glider data and the CTD dataset. We could interpolate over this small data gab but prefer to make clear that two different data

sources are used for this figure. We mention the "white vertical line" now explicitly in the figure caption.

**Figure 7. Delete one "following".**
One "following" has been deleted.

**Figure 9. A depth scale for the study region would help (because the shallow OMZ was studied).**
An approximate depth information was added to the schematic and we further added the information (shallow OMZ and approximate depth scale) to the figure caption.

**Literature:**
Bittig, H. C., B. Fiedler, R. Scholz, G. Krahmann, and A. Koertzinger (2014), Time response of oxygen optodes on profiling platforms and its dependence on flow speed and temperature, Limnol. Oceanogr. Methods, 12(8), 617–636,

Brandt, P., Bange, H. W., Banyte, D., Dengler, M., Didwischus, S.-H., Fischer, T., Greatbatch, R. J., Hahn, J., Kanzow, T., Karstensen, J., Körtzinger, A., Krahmann, G., Schmidtko, S., Stramma, L., Tanhua, T., and Visbeck, M.: On the role of circulation and mixing in the ventilation of oxygen minimum zones with a focus on the eastern tropical North Atlantic, Biogeosciences, 12, 489–512, 2015.

Hahn, J., P. Brandt, R. J. Greatbatch, G. Krahmann, and A. K€ortzinger (2014), Oxygen variance and meridional oxygen supply in the TropicalNorth East Atlantic oxygen minimum zone,Clim. Dyn.,43, 2999–3024

Iversen, M. H., Nowald, N., Ploug, H., Jackson, G. A., and Fischer, G.: High resolution profiles of vertical particulate organic matter export off Cape Blanc, Mauritania: Degradation processes and ballasting effects, Deep Sea Research Part I: Oceanographic Research Papers, 57, 771 – 784, 2010.

Kalvelage, T., Lavik, G., Jensen, M. M., Revsbech, N. P., Löscher, C., Schunck, H., Desai, D. K., Hauss, H., Kiko, R., Holtappels, M., LaRoche, J., Schmitz, R. A., Graco, M. I., and Kuypers, M. M. M.: Aerobic Microbial Respiration In Oceanic Oxygen Minimum Zones, PLOS ONE, 10, 1–17, 2015.

Machu, E., Capet, X., Estrade, P. A., Ndoye, S., Brajard, J., Baurand, F., Lazar, A., and Brehmer, P.: First evidence of Anoxia and Nitrogen Loss in the Southern Canary Upwelling System (Resubmitted), Geophysical Research Letters, 2018.

Thomsen, S., T. Kanzow, G. Krahmann, R. J. Greatbatch, M. Dengler, and G. Lavik (2016), The formation of a subsurface anticyclonic eddy in the Peru-Chile Undercurrent and its impact on the near-coastal salinity, oxygen, and nutrient distributions, *J. Geophys. Res. Oceans*, 121, 476–501, doi: 10.1002/2015JC010878.

---

## Author Comment (AC3) · 27 Nov 2018

Dear Editor, dear Referees,

in the supplement you can find the authors' final response to reviewer 1 as a *.pdf.

On behalf of the authors kind regards, - Soeren Thomsen

Please also note the supplement to this comment:
https://www.biogeosciences-discuss.net/bg-2018-252/bg-2018-252-AC3-supplement.pdf

---

## Referee Report (RR1)

**Second revision of "Remote and local drivers of oxygen and nitrate variability in the shallow oxygen minimum zone off Mauritania in June 2014" by Soeren Thomsen et al.**

**General Assesmemnt**

The authors have successfully addressed my comments/concerns, and they have modified the manuscript accordingly or, at least, they provided convincing explanations of why not doing so. I can now support publication. However, I still have a number of minor concerns and small corrections to the manuscript that must be might before final publication.

**Minor comments and technical corrections**

- P2 – L16: suggests → suggest

- P2 – L28-29: "... an equatorward coastal jet whereas a surface intensified poleward flow with velocities well above 30 cm/s were* observed ..." → was

  P5 – L2-3: "These analyses include the results of the local AOU and OMP analysis in regard to remote (transport) and local respiration and remineralization processes.". The construction of this sentence sounds a bit odd to me, you might want to consider rephrasing it, maybe: "These analyses include the results of the local AOU and OMP analysis in regard to remote (transport) and local (respiration and remineralization) processes."

- P6 – L22-25: The description of the ADCP dataset is a bit confusing. It is not clear from this description how many instruments have been used. From this paragraph it seems that more than one instrument has been used, but this is not coherent with Fig. 4 caption. Please revise this paragraph.

- P6 – L31. Add units to the expression ($r = 1.8417 \cdot ESD^{1.8}$)

- P7 – L5. "(Kalvelage et al., 2015)" → Kalvelage et al. (2015).

- In P. 8 – L22-23 you state "In practice we fitted a **4th order polynomial function** to a group of 18 offshore AOU reference profiles using density as the independent variable" but in L27-28 you refer to "Moreover a smooth offshore AOU ($\sigma$) profile was 30 needed that allow for fitting a **simple linear fit**". Am I missing something or this is contradictory?

- P9 – L 10: "(here first guess Redfield ratio following (Karstensen and Tomczak, 1998), 8.625 = 138 $O_2$ / 16 $\sum NO_x$ )" → revise the use of brackets in this sentence

- P9 – L17: "Mixing of water masses are* not resolved" → is (?)

- P9 – L23: "respired /remineralized signal" → respiration/remineralization signal

- P9 – L24: "were* regional remineralization" → where

- P10 – L4: "relatively cold waters": consider writing "relatively cold surface waters" for more clarity

- P12 – L7: Delete "(not shown)" referring to the mixed layer depth, it is now shown in Fig. 3

- P13 – L20: "The along-shore circulation exhibited elevated variability during June 2014, which are* described based on two ship-based acoustic Doppler current profiler transects" → is

- Fig. 4 and associated description/discussion. Just for my curiosity, are the temporal changes in meridional velocity associated with changes in the wind forcing, while going towards the upwelling relaxation season? If the authors have any information about the origins of this variability, it could be useful to discuss it in the manuscript in relation to the increase in DO at the shelf break. Also, authors could add isopycnals to this figure to illustrate changes in the water mass distribution associated with the offshore shift of the poleward flow. This is just a suggestion.

- Fig 9 caption. Please specify what filled areas represent in this figure (± 1SD, 95% confidence intervals. . . ) P23 – L1: "Cape Blance" → "Cape Blanc"

- P23 – L19: "Its difficult" → "It is difficult"

- P23 – L27: ". . . as the once presented .." → "as those presented"

---

## Author Response (AR2)

Author response (AR1) by Soeren Thomsen for:

Second revision of "Remote and local drivers of oxygen and nitrate variability in the shallow oxygen minimum zone off Mauritania in June 2014" by Soeren Thomsen et al.

**Anonymous Referee #1**
**General Assesmemnt**
**The authors have successfully addressed my comments/concerns, and they have modified the manuscript accordingly or, at least, they provided convincing explanations of why not doing so. I can now support publication. However, I still have a number of minor concerns and small corrections to the manuscript that must be might before final publication.**

> We thank Reviewer #1 for taking the time to go through the full manuscript again in detail and for the help to further improve its quality. We are glad that our in depth revision satisfied the reviewer.

**Minor comments and technical corrections**
**P2 – L16: suggests ! suggest**
> Changed.

**P2 – L28-29: ". . . an equatorward coastal jet whereas a surface intensified poleward flow with velocities well above 30 cm/s were\* observed ..." ! was**
> Changed.

**P5 – L2-3: "These analyses include the results of the local AOU and OMP analysis in regard to remote (transport) and local respiration and remineralization processes.". The construction of this sentence sounds a bit odd to me, you might want to consider rephrasing it, maybe: "These analyses include the results of the local AOU and OMP analysis in regard to remote (transport) and local (respiration and remineralization) processes."**
> Changed. Thanks for the suggestion.

**P6 – L22-25: The description of the ADCP dataset is a bit confusing. It is not clear from this description how many instruments have been used. From this paragraph it seems that more than one instrument has been used, but this is not coherent with Fig. 4 caption. Please revise this paragraph.**
> We revised the paragraph to be more clear on this.

**P6 – L31. Add units to the expression (r = 1:8417 $ESD_{1:8}$)**
> Added.

**P7 – L5. "(Kalvelage et al., 2015)" ! Kalvelage et al. (2015).**
> Changed.

**In P. 8 – L22-23 you state "In practice we fitted a 4th order polynomial function to a group of 18 offshore AOU reference profiles using density as the independent variable" but in L27-28 you refer to "Moreover a smooth offshore AOU () profile was 30 needed that allow for fitting a simple linear fit". Am I missing something or this is contradictory?**
> Thanks for pointing to this error. Indeed we do a 4th order fit and no linear fit. We changed the end of the sentence by and replaced linear fit with reasonable fit.

**P9 – L 10: "(here first guess Redfield ratio following (Karstensen and Tomczak, 1998), 8.625 = 138 $O_2$ / 16 SUM($NO_x$) )" ! revise the use of brackets in this sentence**
> We revised the sentence.

**P9 – L17: "Mixing of water masses are* not resolved" ! is (?)**
    Changed.

**P9 – L23: "respired /remineralized signal" ! respiration/remineralization signal**
    Changed.

**P9 – L24: "were* regional remineralization" ! where**
    Changed.

**P10 – L4: "relatively cold waters": consider writing "relatively cold surface waters" for more clarity**
    Changed.

**P12 – L7: Delete "(not shown)" referring to the mixed layer depth, it is now shown in Fig. 3**
    Thanks for checking so carefully. Deleted.

**P13 – L20: "The along-shore circulation exhibited elevated variability during June 2014, which are* described based on two ship-based acoustic Doppler current profiler transects" ! Is**
    Changed.

**Fig. 4 and associated description/discussion. Just for my curiosity, are the temporal changes in meridional velocity associated with changes in the wind forcing, while going towards the upwelling relaxation season? If the authors have any information about the origins of this variability, it could be useful to discuss it in the manuscript in relation to the increase in DO at the shelf break. Also, authors could add isopycnals to this figure to illustrate changes in the water mass distribution associated with the offshore shift of the poleward flow. This is just a suggestion.**

    We thank the reviewer for this important comment. We had a look on wind speed and the wind stress curl along the 18ºN transect using daily ASCAT (Advanced SCATterometer) wind measurements (see review Fig. 1). The wind speed (black line) shows great variability during June 2014 and ranges from 2.5 m/s to 10 m/s (Fig. 1, left panel). However, in our view no clear trend is visible. The wind stress curl shows a slight increase during the second half of June (Fig. 1, right panel).

[Figure]

    Figure 1: Zonal (blue), meridional (red) and absolute (black) wind speed averaged the 18ºN transects (left panel). Zonal (blue) and meridional wind stress curl components (red) and absolute wind stress curl (black) averaged along the 18ºN transects (left panel).

We added a figure to the response letter to show the changes in the wind speed and wind stress curl. This allows the reviewer and interested readers to see the local changes in wind forcing. However we doubt that it is possible to relate the changes in the velocity field on such short time and space scales to the highly variable local wind forcing. This is due to the importance of remote forcing (Kounta et al. 2018), adjustment time scales and because the region is characterised by strong mesoscale variability. It is beyond the scope of this study to fully investigate and disentangle the drivers of the observed changes in the meridional velocity, if ever this is possible with the observations at hand. A proper analysis would require an own full manuscript. However, we now added a short discussion on this circulation changes and refer to the modelling study of Kounta et al. (2018). For our study the offshore movement of the Undercurrent is of mayor importance as it leads to a reduction of the alongshore velocities on the upper shelf. This offshore displacement might be related to the offshore propagation of Rossby waves as described in Kounta et al. (2018).

Isopycnals are now also added to the figure.

**Fig 9 caption. Please specify what filled areas represent in this figure ( 1SD, 95% confidence intervals. . . )**
Done.

**P23 – L1: "Cape Blance" ! "Cape Blanc"**
Changed.

**P23 – L19: "Its difficult" ! "It is difficult"**
Changed.

**P23 – L27: ". . . as the once presented .." ! "as those presented"**
Changed.

—————————————————————————————————————————————

**Author reply to final comments by Associate Editor Marilaure Grégoire**
We thank the associate editor Marilaure Grégoire for her final comments which further improve the quality of our manuscript.

**Minor comments**
**Page 1, line 18 : Shadow or shallow ?**
Shadow zone (see Luyten et al., 1983).

**Page 2 : please define ITCZ**
Done. We write out ITCZ now.
**Page 3, lines 12-13 : « Besides the local water column remineralization and respiration the benthic oxygen uptake » I would add a coma between "respiration" and "the benthic oxygen uptake".**
Added.

**Page 7, lines 5 and 9: Kalvelage et al., (2015) and not (Kalvelage et al. 2015)**
Changed.

**Page 8, line 11: describe and not described**
Corrected.

**Page 8, line 23: I do not understand the reference to figure 1b here.**
In figure 1b we show the position off the reference profiles with red circles.

**Page 9, line 24: where and not were**
    Changed.

**Page 13: usually fluorescence is measure in rfu while chlorophyll is indeed a concentration. I guess that here you mean the chlorophyll concentration estimated from fluorescence measurement. Is it correct? Please clarify.**

    Yes this is correct. We are referring to chlorophyll concentrations. Thanks for this comment. We have corrected this throughout the whole manuscript and also added one additional line in the data and method section to make this more clear.

**Page 20, Fig. 8 and 9: is there any reason why the DOC concentration is expressed per volume of water while oxygen concentration is expressed per kg? Please, justify.**

    Indeed there is no clear reason why the DOC and AA concentrations were expressed per volume beside "community traditional" reasons. However we now express both DOC and AA in per kg to be more coherent. Additionally per kg has the clear advantages that it is conservative. The new unit results in about 2.5 to 2.7% "smaller" values for DOC and AA.

**Page 26: simulations**
    Corrected.

**Figure 1, legend, last line: "are shown" instead of "is shown".**
    Corrected.

**Figure 3: Is it chlorophyll fluorescence in $\mu g/l$ or concentration? Mixed layer depth (-0.2°), please clarify.**

    As mentioned above we refer to chlorophyll concentrations.

    Our mixed layer depth is defined by a typically used mixed layer temperature minus $0.2^oC$ criterium. So at the depth were the temperature is $0.2^oC$ below the surface temperature we find our mixed layer depth. We made this clearer now in the figure caption.

**Figure 4: please specify the positive direction**
    Positive means northward flow. We added this information to the figure caption.

[revised manuscript text omitted]